# Offshore wind farm global blockage measured with scanning lidar

Jörge Schneemann[1,*], Frauke Theuer[1,*], Andreas Rott[1], Martin Dörenkämper[2], and Martin Kühn[1]

[1]ForWind, Institute of Physics, Carl von Ossietzky University Oldenburg, Küpkersweg 70, 26129 Oldenburg, Germany
[2]Fraunhofer Institute for Wind Energy Systems, Küpkersweg 70, 26129 Oldenburg, Germany
[*]These authors contributed equally to this work.

**Correspondence:** Jörge Schneemann (j.schneemann@uol.de)

**Abstract.** The objective of this paper was the experimental investigation of the accumulated induction effect of a large offshore wind farm as a whole, i.e. the global blockage effect, in relation to atmospheric stability estimates and wind farm operational states. We measured the inflow of a $400\,\mathrm{MW}$ offshore wind farm in the German North Sea with a scanning long-range Doppler wind lidar. A methodology to reduce the statistical variability of different lidar scans at comparable measurement conditions was introduced and an extensive uncertainty assessment of the averaged wind fields was performed to be able to identify the global blockage effect which is small compared to e.g. wind turbine wake effects and ambient variations in the inflow. Our results showed a $4\,\%$ decrease in wind speed (accuracy range $2\,\%$ to $6\,\%$) at transition piece height ($24.6\,\mathrm{m}$) upwind of the wind farm with the turbines operating at high thrust coefficients above $0.8$ in a stably stratified atmosphere, which we interpreted as global blockage. In contrast, at unstable stratification and similar operating conditions and for situations with low thrust coefficients (i.e. approx. $0$ for not operating turbines and $\leq 0.3$ for turbines operating far above rated wind speed) we identified no wind speed deficit. We discussed the significance of our measurements, possible sources of error in long-range scanning lidar campaigns and give recommendations how to measure small flow effects like global blockage with scanning Doppler lidar. In conclusion, we provide strong evidence for the existence of global blockage in large offshore wind farms in stable stratification and the turbines operating at a high thrust coefficient by planar lidar wind field measurements. We further conclude that global blockage is dependant on atmospheric stratification.

## 1 Introduction

Wind turbine wakes can cause negative effects at downstream turbines due to decreased wind speeds and increased turbulence (Porté-Agel et al., 2019). This was intensively studied in the last decades and is considered in all wind farm projects planned today (Rohrig et al., 2019). Recently, the so-called global blockage effect came into the research focus. It denotes the reduction of the wind speed in a comparably wide area upstream of large wind farms. The effect is supposed to be caused by an interaction of the wind farm as a whole with the atmospheric boundary layer since it can not be sufficiently described by a simple superposition of the induction zones of individual turbines in a large wind farm. Global blockage is usually not considered in the planning of wind energy projects and could therefore lead to a non-negligible bias in the assessment of the wind resource (Bleeg et al., 2018).

The knowledge of the wind resource to be expected during the lifetime of a wind energy project is crucial for its successful financing and economic operation. A large wind farm operator recently attributed a reduction in the predicted unlevered life-cycle Internal Rate of Return (IRR) among others to underestimated wake effects between distant wind farms and the global wind farm blockage effect (Ørsted A/S, 2019).

The induction zone of a single wind turbine describes the region in front of the rotor where the wind speed is reduced due to the presence of the wind turbine. The standard for onshore power curve measurements of wind turbines recommends to measure the free wind speed, i.e. the wind speed at the turbine location in absence of the turbine, at least 2.5 rotor diameters D upstream or lateral to the turbine's location (IEC, 2017). It is assumed that the influence of a wind turbine's induction zone is very low at this distance. The effect of reduced wind speeds in the induction zone of a wind turbine is called wind turbine blockage effect and it is caused by the thrust of the rotor. Meyer Forsting et al. (2017) give an overview of wind turbine blockage and the induction zone.

The accumulated induction zone generated by the wind farm as a whole is called global blockage and leads to a wind speed deceleration and flow deflections sideways and upward in front of the wind farm. As for solid objects in the flow like mountains or buildings a wind farm represents an obstacle causing an upstream reverse pressure gradient which results in reduced wind speeds. Different from solid objects wind farms are porous and actively produce thrust. In case of a wind farm, this reverse pressure gradient is referred to as global blockage. Wind farm related factors influencing the extent and the intensity of the global blockage effect are wind farm size, layout, wind direction, turbine spacing and thrust coefficient (Porté-Agel et al., 2019). A meteorological parameter that affects the extent and strength of the wind farm induction zone, i.e. the global blockage, is the height of the atmospheric boundary layer (Porté-Agel et al., 2019) which is related to atmospheric stability (Kitaigorodskii and Joffre, 1988).

Knowledge about the global blockage effect mainly results from numerical studies. Meyer Forsting et al. (2016) used RANS simulations to investigate the effect that wind turbines in a row have on each other's power production when considering different inflow directions. They found a combined induction zone of the whole turbine row with changes in the individual turbine's power in the range of -1 % to 2 % while the accumulated power remained nearly constant. Wu and Porté-Agel (2017) performed Large Eddy Simulations (LES) of large finite-size wind farms in neutral stratified boundary layers capped with a thermally-stratified free atmosphere (conventionally-neutral atmospheric boundary layer) and discovered a wind farm induction zone extending about 0.8 km (rotor diameter $D$=80 m ) upwind and leading to power reductions of 1.3 % and 3 % for different farm layouts. Using LES, Allaerts and Meyers (2017) determined wind farms to excite gravity waves in stable stratification which are caused by the upward movement of the top of the boundary-layer due to global wind farm blockage. Those gravity waves are similar to so called mountain waves induced by hills and mountains.

Some modelling studies analyzed global blockage from implementations of the accumulated turbine induction in engineering models. Branlard and Meyer Forsting (2020) introduced a computational inexpensive vortex model to assess wind farm production considering accumulated blockage effects. The wind turbine and wind farm blockage effects resolved with their model compared well with results from actuator disk simulations at moderate thrust coefficients. Bleeg (2020) used a graph neural network surrogate model to predict wind turbine interaction losses including global blockage. He found a good agreement of

the model and the results of RANS simulations. Nygaard et al. (2020) coupled an engineering model for global blockage with a wind turbine wake model modified to better represent the far wind farm or cluster wake. Their wind farm blockage model was able to predict the trend in the variation of power in the front row of turbines but underestimated its amplitude. The authors pinpoint that more research is needed on the further model development and calibration. Branlard et al. (2020) presented a current overview of engineering models including global blockage and compare their performance with an actuator disk RANS simulation as reference. They find the different models to show varying levels of accuracy with a mean error level below 1 % in the induction zone.

These numerical studies agree on the magnitude of the wind speed deficit in the wind farm induction zone to be in a lower one-digit percentage range. Nevertheless, most numerical studies lack measurement data for validation since experimental investigations on global blockage have been rarely performed.

Segalini and Dahlberg (2019) measured the effect of a model wind farm on a row of upstream turbines in different distances in a laminar wind tunnel. They observed a decrease in wind speed at the turbine row in distances of up to 30 rotor diameters (D=45 mm) upstream and with a maximum of 2 %.

To our knowledge the only study presenting free field measurements of global blockage was performed by Bleeg et al. (2018), who analysed wind measurements of meteorological masts upstream and lateral to three different onshore wind farms before and after the commercial operation date and for high thrust coefficients of the turbines. Deficits in wind speed upstream compared to the lateral reference mast of around 2 % and up to more than 6 % appeared typically in front of the farms after the turbines went into operation. The authors relate this mainly to the global blockage effect.

Open field measurements of global blockage are challenging. Classic anemometry is limited in its possibilities to study the induction zone of a wind farm since just a limited number of masts can be placed in front of it due to mainly financial constraints. In the last decade, the remote sensing methods Doppler wind lidar (light detection and ranging) has become a common tool in many fields of wind energy research and applications (Hasager and Sjöholm, 2019). Lidar devices offer the possibility to scan whole wind fields with ranges of several kilometres. Commercial scanning lidar systems allow to measure the line-of-sight (LOS) component of the wind vector on several hundred positions along the emitted laser beam and to orientate the beam in any direction. Scanning lidars have enabled many new insights in different fields of wind energy research, like wind turbine wakes (Käsler et al., 2010; Trabucchi et al., 2014), wind farm cluster wakes (Schneemann et al., 2020), resource assessment in complex terrain (Menke et al., 2020) and minute-scale wind power forecasts (Theuer et al., 2020b).

The current knowledge on global blockage is mainly based on modelling activities or wind tunnel studies. Compared to well-known phenomena like wind turbine wakes with significant wind speed deficits in the order of tens of percents of the inflow wind speed in a well defined downstream region, global blockage is much harder to study especially due to the larger spatial expansion over typically several square kilometres and the smaller wind speed differences in a single digit percentage range. Furthermore, the effect of global blockage needs to be separated from other spatial and temporal variations in the wind field. Averaged field measurements on a small number of scattered points (Bleeg et al., 2018) lack information on superposed

flow features like local wind speed variations due to orography, wind farm layout or varying meteorological conditions. Since it is not possible to distinguish these flow features, this adds uncertainty to the identification of global blockage.

Therefore, accurate field measurements spatially resolving the induction zone of the wind farm are of major importance to validate the modelling results already achieved. The extent of global blockage in operating wind farms and its dependency on 100 atmospheric stability and the farm's operational state is still not fully understood and proof for the effect appearing in operating wind farms is still missing.

Compact Doppler lidar systems offer the possibility to scan large parts of the inflow of a wind farm with measurement ranges up to 10 kilometres. Nevertheless, to obtain wind data for a quantitative analysis all measurement parameters of the lidar device such as its orientation and tilt due to platform movements need to be carefully selected and accurately controlled. Furthermore, 105 environmental parameters and conditions like curvature of the earth, knowledge of the current wind profile and atmospheric stability for height correction need to be known and accounted for.

The objective of our paper is the experimental assessment of the global blockage effect in a large offshore wind farm dependent on atmospheric stability estimates and wind farm operational states. In addition to this, we are proposing a method 110 to examine comparably small flow effects like global blockage with long-range scanning lidar. Our approach includes:

– analysing horizontal long-range Doppler lidar plan position indicator (PPI) scans upstream of a 400 MW offshore wind farm,

– deriving atmospheric stability from local meteorological measurements and

– performing a detailed uncertainty assessment and error correction on all measured quantities.

Furthermore, we provide recommendations for measurements of global blockage or similar small flow effects with scanning Doppler Lidar.

In this paper we use the terms *blockage effect* and *wind turbine blockage effect* for decreased wind speeds in the induction zone of single turbines while we call the accumulated blockage effect of all turbines within a wind farm or wind farm cluster, i.e. the reverse upstream pressure gradient of the wind farm, *global (wind farm) blockage* or *global (wind farm) blockage effect*. 120 This paper is structured as follows. In Section 2 we introduce our experimental setup including lidar measurements and atmospheric stability estimations. We place special focus on the uncertainty assessment of the lidar data. We present the results of four different inflow situations varying in atmospheric stability and wind speed in Section 3. In Section 4 we discuss our findings and give recommendations for lidar measurements of flow effects like global blockage with a magnitude of typical ambient wind speed fluctuations. We conclude and close the paper in Section 5.

## 2 Methods

In this section we describe the analyzed wind farm (Section 2.1), lidar measurements (Section 2.2), meteorological measurements and atmospheric stability characterization (Section 2.3), lidar data analysis (Section 2.4) and lidar wind speed uncertainty estimation (Section 2.5).

### 2.1 Offshore wind farm Global Tech I

At the time of our measurement campaign between August 2018 and January 2020, several offshore wind farms had been installed in the German and Dutch North Sea. In the focus of this work is the 400 MW wind farm «Global Tech I» (GT I). It features 80 turbines of type «Adwen AD 5-116» with a rotor diameter $D$ of 116 m, a hub height of 92 m, a rated power of 5 MW at a rated wind speed of 12.5 ms$^{-1}$ and a cut-in wind speed of 4 ms$^{-1}$ (Global Tech I Offshore Wind GmbH, 2021). Figure 1 (a) gives an overview of the region around GT I while Figure 1 (b) displays its layout. Supervisory Control and Data Acquisition (SCADA) data of GT I were available to check the turbines status and the power production within regarded lidar scan intervals.

The «BorWin 1» cluster consisting of the operating wind farms «Veja Mate» and «BARD Offshore 1» and the wind farm «Deutsche Bucht» currently under construction is located in a distance of about 24 km in south-west direction (green shapes in Figure 1 (a)). During our measurements the wind farms «Hohe See» and «Albatros» (open blue shapes in Figure 1 (a) and turbine coordinates in Figure 1 (b)) were build in the direct south-west vicinity of GT I within approx. 1 to 6 km distance upstream with several transition pieces, turbines and two sub-stations installed. Measurements after the first power was fed in on 15 July 2019 were not considered (EnBW, 2019). Hard targets in the lidar data due to the construction of the two upstream wind farms «Hohe See» and «Albatros» were filtered from the measurements (cf. Section 2.4) and can lead to a reduced data availability on the corresponding line of sight direction due to (partial) laser beam coverage.

### 2.2 Lidar measurements

We used a scanning long-range Doppler wind lidar of type Leosphere Windcube 200S (Serial no. WLS200S-024) which we installed on the transition piece (TP, platform to access the turbine) of turbine GT58 in GT I (red filled ⋄ in Figure 1 (b)). A photograph of the lidar on the TP is provided by Schneemann et al. (2020), Figure 1. The height of the lidar's scanner was approximately 24.6 m above mean sea level (MSL), 67.0 m below hub height and 9.0 m below lower blade tip height of the turbine. The measurement campaign started in August 2018 and ended in January 2020. We consider data from a period between February 2019 and June 2019. We performed plan position indicator scans (PPI) with an elevation of $0°$, resulting in a measurement height of 24.6 m MSL plus a correction due to the earth's curvature (up to 5 m in 8 km distance). Further, a turbine thrust-dependent tilt of the lidar was observed, resulting in varying measuring heights across range gates and azimuth angles (cf. Section 2.5). We set the lidar's pulse length to 400 ns, the acquisition time to 2.0 s, the scanning speed to $1° s^{-1}$ and scanned the upstream flow in different azimuth sectors of $150°$ that we aligned manually to the wind direction. Range gates were defined between 500 m (approx. 4 $D$) and 7990 m (approx. 69 $D$) with a 35 m spacing. After intensive data filtering

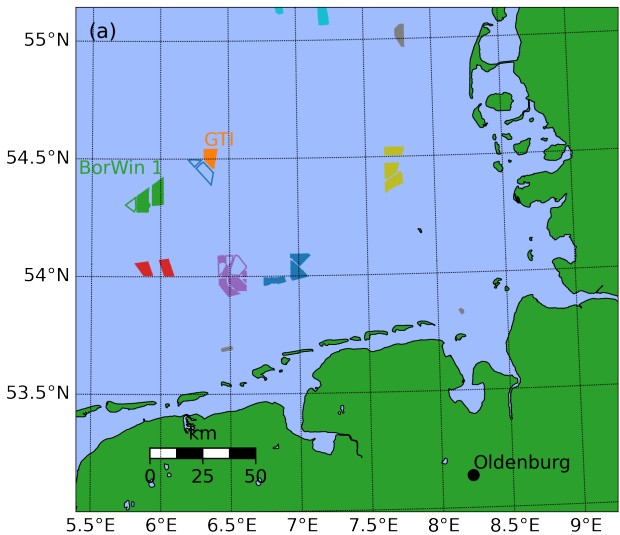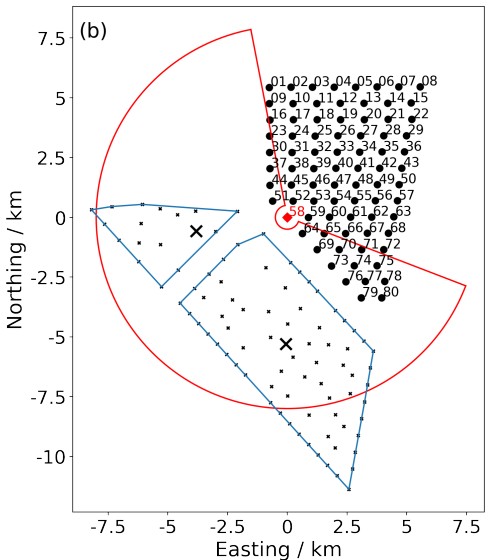

**Figure 1.** (a) Location of the wind farm GT I (orange) in the North Sea with neighbouring wind farms and clusters shown. Wind farms under construction are depicted as open shapes (status beginning of 2019). The 24 km upstream «BorWin 1» cluster is marked in green. (b) Layout of GT I (● with turbine numbers). The position of the lidar on turbine GT58 (red filled ◇) and the lidar scan region covering the area of the four different $150°$ scan sectors (red line) are highlighted. The turbine locations (small ×) and the substations (×) of the wind farms «Hohe See» (southerly blue shape) and «Albatros» (northerly blue shape), which were under construction during our measurements, are marked.

typical ranges around 5500 m were achieved (cf. Section 2.4). One lidar scan took 150 s and resulted in 215 range gates (also referred to as "measurement points") on each of the 75 beams. The further processing of the lidar scans is described in Section 2.4. Schneemann et al. (2020) give further information on the measurements and Schneemann et al. (2019) provide
some exemplary lidar scans from this campaign.

### 2.3   Atmospheric stability characterization and meteorological measurements

For the analysis of the global blockage effect knowledge about wind speed at one common height across the whole scan is required. The varying measuring height, as a consequence of the tilt of the lidar device and the Earth's curvature, thus necessitate the extrapolation of wind speeds to that common height. To keep extrapolation distances small, we here chose
the height of the transition piece/lidar device. For the extrapolation we use a logarithmic stability corrected wind profile. Information regarding stability further allows us to analyse the effect of atmospheric conditions on global blockage.

We used a similar methodology to derive atmospheric stratification as in Theuer et al. (2020b) and Schneemann et al. (2020), which is described here for completeness. To characterize atmospheric stability (Emeis, 2018) we used local measurements as well as reanalysis data. On the transition piece of turbine GT58 close to the lidar's position, we measured air temperature and
humidity (Vaisala HMP155) and air pressure (Vaisala PTB330). Additionally, we used the sea surface temperature (SST) from

the OSTIA data set (Good et al., 2020). We utilized a methodology introduced by Rodrigo et al. (2015) to estimate the Bulk Richardson number

$$Ri_{\mathrm{b}} = \frac{g}{T_{\mathrm{v}}} \frac{0.5\, z_{\mathrm{TP}}\, (\Theta_{\mathrm{TP}} - \Theta_0)}{u_{\mathrm{li}}^2}. \tag{1}$$

Here, $g$ is the gravitational acceleration, $T_{\mathrm{v}}$ the virtual temperature at sea level and $\Theta_{\mathrm{TP}}$ and $\Theta_0$ the virtual potential temperature at TP height $z_{\mathrm{TP}}$ and sea level respectively. $u_{\mathrm{li}}$ describes the wind speed at the lidar position, determined utilising lidar measurements up to range gates of $600\,\mathrm{m}$. The height used to calculate $Ri_{\mathrm{b}}$ is defined as the mean between the two height levels, i.e. $0.5 z_{\mathrm{TP}}$. After estimating $Ri_{\mathrm{b}}$ we obtain the dimensionless stability parameter

$$\zeta = \begin{cases} \frac{10 Ri_{\mathrm{b}}}{1 - 5 Ri_{\mathrm{b}}} & Ri_{\mathrm{b}} > 0 \\ 10 Ri_{\mathrm{b}} & Ri_{\mathrm{b}} \leq 0 \end{cases} \tag{2}$$

and finally the Obukhov length

$$L = \frac{0.5\, z_{\mathrm{TP}}}{\zeta}. \tag{3}$$

We estimated the roughness length $z_0$ using the determined Obukhov length $L$ and the stability corrected logarithmic wind profile

$$u = \sqrt{\frac{z_0\, g}{\alpha_c}} \frac{1}{\kappa} \left( \ln\left(\frac{z}{z_0}\right) - \Psi\left(\frac{z}{L}\right) \right), \tag{4}$$

with $z = z_{\mathrm{TP}}$ and $u = u_{\mathrm{TP}}$. Here, $\kappa = 0.4$ describes the von Kármán-constant and $\alpha_c = 0.011$ the Charnock parameter, often used in an offshore context (Smith, 1980). The stability correction term

$$\Psi = \begin{cases} 2\ln\left(\frac{1+x}{2}\right) + \ln\left(\frac{1+x^2}{2}\right) - 2\arctan(x) + \frac{\pi}{2} & L < 0, \text{ where } x = (1 - \gamma \frac{z}{L})^{1/4} \\ -\beta \frac{z}{L} & L \geq 0 \end{cases} \tag{5}$$

was defined following Dyer (1974) with $\gamma = 19.3$ and $\beta = 6$ (Högström, 1988).

## 2.4 Lidar data processing

We filtered the lidar scans using a carrier-to-noise (CNR) threshold filter, considering only values with $-26\,\mathrm{dB} < \mathrm{CNR} < 0\,\mathrm{dB}$. With a Velocity-Azimuth-Display (VAD) algorithm, we calculated a mean wind speed $\overline{u}$ and wind direction $\chi$ individually for each scan assuming a homogeneous wind field and neglecting the vertical wind speed component (Werner, 2005). At each measurement point we projected the line-of-sight (LOS) wind velocities $u_{\mathrm{LOS}}$ onto the mean wind direction by means of

$$u_{\mathrm{h}} = \frac{u_{\mathrm{LOS}}}{\cos(\vartheta - \chi)}, \tag{6}$$

with horizontal wind speed $u_{\mathrm{h}}$ and azimuth angles $\vartheta$. Sectors with azimuth angles almost perpendicular to the wind direction, i.e.

$$75° < |\vartheta - \chi| < 105°, \tag{7}$$

were neglected as they are associated with large errors. Further, outliers with $|u_\mathrm{h}-\overline{u}| > 2.75\,\sigma_\mathrm{u}$, with $\sigma_\mathrm{u}$ defined as the standard deviation of horizontal wind speed within each scan, were discarded. For each measurement point we then determined the measuring height, considering both the curvature of the Earth and a turbine thrust-dependent tilt of the lidar device. Further details about the alignment of the lidar and the correction function to estimate the measurement height are presented in Rott et al. (2020). After assessing the measuring height, wind speed values were extrapolated to the lidar height $z_\mathrm{TP} = 24.6\,\mathrm{m}$ using a stability corrected logarithmic wind speed profile described by Equations 4 and 5 (Section 2.3).

Only scans with a data availability of at least $60\,\%$ were considered for further analysis. Data availability was calculated individually for each scan, including measurement points up to a range gate of $7000\,\mathrm{m}$ and not considering critical sectors as defined in Equation 7.

Finally, we interpolated all valid scans to a Cartesian grid with a spacing of $\Delta x = \Delta y = 50\,\mathrm{m}$ to be able to average data of varying scanning sectors.

For further analysis, the lidar scans were categorised according to their respective mean wind direction $\chi$, 10-minute-mean wind speed measured by a sonic anemometer at the nacelle of turbine GT58 $\overline{u}_\mathrm{GT58}$ and atmospheric stability characterized by $L$. In each category, consisting of $N$ individual lidar scans $i$, we performed the following steps: First, the mean wind speed within the scan $i$ at TP height $\overline{u}_{\mathrm{TP},i}$ was derived and used to normalise the wind speeds on all grid points, yielding $u_{\mathrm{TP,norm},i}$. Second, all normalised scans were averaged to $\overline{u}_\mathrm{TP,norm}$. Hereby, Cartesian grid points with data availability $<80\,\%$, i.e. $N_r < 0.8N$, with $N$ the number of all available scans within the category and $N_r$ the number of valid scans at each grid point, were neglected. Third, normalised and averaged wind speeds $\overline{u}_\mathrm{TP,norm}$ were interpolated onto a virtual line in mean wind direction upstream of the lidar.

For the blockage analysis we decided to distinguish between two stability classes, i.e. unstable and stable situations, and three different operational states respectively wind speed ranges. The operational state of the wind farm was estimated using SCADA power data and the wind speed range based on the wind speed at nacelle height. These states are namely the wind farm not operating (below cut-in wind speed, low thrust coefficient of approx. 0), the wind farm operating at rated power (above rated wind speed, low to moderate thrust coefficient) and the wind farm operating below and up to rated power (below rated wind speed, high thrust coefficient). In total, the combination of these two categories left us with a number of six possible cases to be analysed as summarized in Table 1.

**Table 1.** Summary of possible scenarios to be analysed. Those shown in this work are named, those not shown are marked as $X$.

|  | unstable ($L < 0$ m and $|L| < 1000$ m) | stable ($L > 0$ m and $|L| < 1000$ m) |
| --- | :---: | :---: |
| not operating | $X$ | Scenario 2, Figure 5 |
| operating below rated wind speed | Scenario 1, Figure 4 | Scenario 4, Figure 7 |
| operating above rated wind speed | $X$ | Scenario 3, Figure 6 |

However, for brevity we omitted the combinations *unstable, not operating* and *unstable, operating above rated wind speed*. With the comparison of the four remaining cases we aimed to cover both scenarios where global blockage is likely to occur and those where an occurrence is less likely. This "cross-check" allowed us to better interpret the obtained results in terms of possible wind speed gradients caused by other background phenomena. We start with the analysis of the remaining unstable scenario and then continue with the stable cases, sorted according to their thrust coefficients. The four scenarios are summa-

rized below.

**Scenario 1:** Wind turbines operating below and up to rated power with a high thrust coefficient $> 0.8$ at the plateau-region of the thrust-coefficient curve. We chose a wind speed interval of $8\,\mathrm{ms}^{-1} < \overline{u} < 11\,\mathrm{ms}^{-1}$, unstable atmospheric conditions and a total power production of the wind farm with at least $50\,\%$ of wind farm's rated power and $80\,\%$ of the wind farm's estimated

power. Here, the wind farm power is estimated by extrapolating $u_{\mathrm{TP}}$ to hub height using an average logarithmic profile (see Section 2.3, with $L = -300\,\mathrm{m}$) and transferring the result to the whole wind farm considering wind farm effects. Further, only situations with high power production at GT58 ($P_{\mathrm{GT58}} \geq 4000\,\mathrm{kW}$) were considered to make the experienced tilt of the lidar device comparable to that of Scenario 4.

**Scenario 2:** Wind farm not operating with wind speeds below cut-in wind speed, i.e. thrust coefficients of approx. 0. Here scans with wind speed $u_{\mathrm{TP}}$ from $3\,\mathrm{ms}^{-1} < \overline{u} < 4\,\mathrm{ms}^{-1}$, during stable atmospheric conditions and a wind farm power production $< 5\,\%$ of the wind farm's rated power were selected.

**Scenario 3:** Wind farm operating at rated power with wind speeds $u_{\mathrm{TP}}$ above rated wind speed and low thrust coefficients

$\leq 0.3$. This comprises scans with $16\,\mathrm{ms}^{-1} < \overline{u} < 22\,\mathrm{ms}^{-1}$, stable atmospheric conditions and a total wind farm power $> 80\,\%$ of the rated power. Further, only cases with a blade pitch from SCADA data $> 5°$ at GT58 were considered.

**Scenario 4:** Wind turbines operating below and up to rated power with a high thrust coefficient $> 0.8$. Same as Scenario 1, however here we chose scans with stable atmospheric conditions. In this case the estimated wind farm power was determined

using an average logarithmic profile with $L = 300\,\mathrm{m}$.

## 2.5 Uncertainty estimation

For the further analysis and interpretation of the results, several uncertainties introduced in the course of the measurement campaign and data analysis procedure are important to consider. In this section, we qualitatively summarise the most important error contributions and subsequently estimate uncertainties using three different methodologies. First, we calculate the total

255 propagated uncertainty using the uncertainties assigned to the individual components with Gaussian error propagation, second we determine the total propagated uncertainty as before but distinguish also between range gate-independent and range gate-correlated input variables, and third we derive the statistical standard error of the mean.

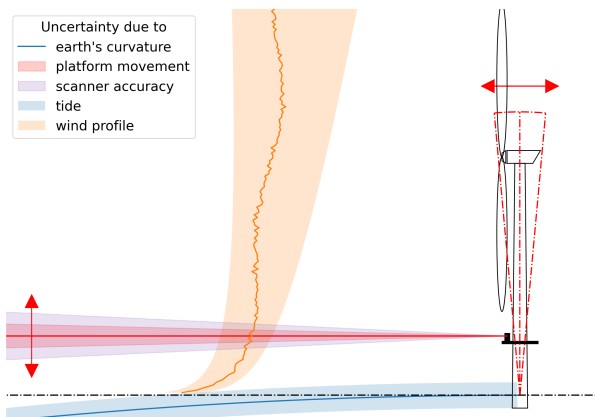

**Figure 2.** Illustration of different sources of uncertainty for wind speed estimates in long-range Doppler lidar measurements on an offshore platform like the TP of an offshore wind turbine. Aside the general uncertainty in the LOS wind speed measurement the main source of uncertainty is the varying measurement height due to lidar scanner misalignment (purple) and platform tilts and movements (red) e.g. due to the turbine's thrust. Curvature of the earth (blue) and tide (light blue) adds on the height uncertainty. As a consequence of known height errors measured wind speeds need to be transformed back to the desired height, thus the lack of knowledge of the prevailing wind profile introduces additional uncertainty (orange).

We summarise sources of errors and uncertainties that need to be considered in offshore lidar measurements of flow effects with small deviations with respect to the mean flow in Table 2 and visualise them in Figure 2. It becomes clear that several of the error sources are directly or indirectly linked to the alignment of the lidar: The device's tilt causes the need for a height extrapolation, thus wind profile information is required, introducing additional uncertainties. Considering a measurement scenario with perfect horizontal measurements, the error sources could be significantly reduced. However, as in this set-up an extrapolation of wind speed $u_{\mathrm{m}}$ at measuring height $z_{\mathrm{m}}$ to lidar height $z_{\mathrm{TP}}$ is required we estimated the uncertainty associated with it in more detail in the following.

### 2.5.1 Total propagated uncertainty

As stated earlier the height extrapolation of lidar data is performed by means of a stability corrected logarithmic wind speed profile (Equation 4). The wind speed at height of the TP $u_{\mathrm{TP}}$ can thus be expressed as

$$u_{\mathrm{TP}} = u_{\mathrm{m}} \frac{\ln\left(\frac{z_{\mathrm{TP}}}{z_0}\right) - \Psi\left(\frac{z_{\mathrm{TP}}}{L}\right)}{\ln\left(\frac{z_{\mathrm{m}}}{z_0}\right) - \Psi\left(\frac{z_{\mathrm{m}}}{L}\right)}. \tag{8}$$

Gaussian error propagation yields the total propagated uncertainty

$$
\Delta u_{\mathrm{TP}} = \left[ \left( \frac{\ln(\frac{z_{\mathrm{TP}}}{z_0}) - \Psi_{\mathrm{TP}}}{\ln(\frac{z_{\mathrm{m}}}{z_0}) - \Psi_{\mathrm{m}}} \Delta u_{\mathrm{m}} \right)^2 + \left( \frac{u_{\mathrm{m}}(\ln(\frac{z_{\mathrm{TP}}}{z_0}) - \Psi_{\mathrm{TP}} + \Psi_{\mathrm{m}} - \ln(\frac{z_{\mathrm{m}}}{z_0}))}{z_0(\ln(\frac{z_{\mathrm{m}}}{z_0}) - \Psi_{\mathrm{m}})^2} \Delta z_0 \right)^2 + \left( \frac{u_{\mathrm{m}}(\Psi_{\mathrm{TP}} - \ln(\frac{z_{\mathrm{TP}}}{z_0}))}{z_{\mathrm{m}}(\ln(\frac{z_{\mathrm{m}}}{z_0}) - \Psi_{\mathrm{m}})^2} \Delta z_{\mathrm{m}} \right)^2 \right.
$$

(9)

$$
\left. + \left( \frac{u_{\mathrm{m}}}{\Psi_{\mathrm{m}} - \ln(\frac{z_{\mathrm{m}}}{z_0})} \Delta \Psi_{\mathrm{TP}} \right)^2 + \left( \frac{u_{\mathrm{m}}(\ln(\frac{z_{\mathrm{TP}}}{z_0}) - \Psi_{\mathrm{TP}})}{(\ln(\frac{z_{\mathrm{m}}}{z_0}) - \Psi_{\mathrm{m}})^2} \Delta \Psi_{\mathrm{m}} \right)^2 \right]^{1/2},
$$

with the corresponding uncertainty in the stability correction term

$$
\Delta \Psi = \begin{cases} \left| \frac{4x^2}{x^3 + x^2 + x + 1} \right| \Delta x & L < 0, \\[2mm] & \text{where } \Delta x = \left[ \left( -\frac{\gamma}{4L} \left( 1 - \gamma \frac{z}{L} \right)^{-3/4} \Delta z \right)^2 + \left( \frac{\gamma z}{4L^2} \left( 1 - \gamma \frac{z}{L} \right)^{-3/4} \Delta L \right)^2 \right]^{1/2}. \\[2mm] \left[ \left( -\beta \frac{1}{L} \Delta z \right)^2 + \left( \beta \frac{z}{L^2} \Delta L \right)^2 \right]^{1/2} & L \geq 0 \end{cases}
$$

(10)

The indices of the correction terms $\Psi$ refer to the height at which it is determined. The uncertainty of the Obukhov length $L$ is also determined by means of Gaussian error propagation of Equations 1 to 3, leading to

$$
\Delta Ri_{\mathrm{b}} = \left[ \left( \frac{-g}{T_{\mathrm{v}}^2} \frac{0.5 \, z_{\mathrm{TP}} \, (\Theta_{\mathrm{TP}} - \Theta_0)}{u_{\mathrm{li}}^2} \Delta T_{\mathrm{v}} \right)^2 + \left( \frac{-2g}{T_{\mathrm{v}}} \frac{0.5 \, z_{\mathrm{TP}} \, (\Theta_{\mathrm{TP}} - \Theta_0)}{u_{\mathrm{li}}^3} \Delta u_{\mathrm{li}} \right)^2 + \left( \frac{g}{T_{\mathrm{v}}} \frac{0.5 \, z_{\mathrm{TP}}}{u_{\mathrm{li}}^2} \Delta \Theta_0 \right)^2 \right.
$$

(11)

$$
\left. + \left( \frac{g}{T_{\mathrm{v}}} \frac{0.5 \, z_{\mathrm{TP}}}{u_{\mathrm{li}}^2} \Delta \Theta_{\mathrm{TP}} \right)^2 \right]^{1/2},
$$

(12)

$$
\Delta \zeta = \begin{cases} \left| \frac{10}{(1 - 5Ri_{\mathrm{b}})^2} \right| \Delta Ri_{\mathrm{b}} & Ri_{\mathrm{b}} > 0 \\[2mm] 10 \Delta Ri_{\mathrm{b}} & Ri_{\mathrm{b}} \leq 0, \end{cases}
$$

(13)

$$
\Delta L = \left| \frac{-0.5 \, z_{\mathrm{TP}}}{\zeta^2} \right| \Delta \zeta.
$$

(14)

The uncertainties $\Delta T_v$, $\Delta \Theta_0$ und $\Delta \Theta_{\mathrm{TP}}$ are hereby assessed using air and water temperature, humidity and pressure uncertainties. We set $\Delta T_{\mathrm{air}} = 0.1\,\mathrm{K}$, $\Delta T_{\mathrm{water}} = 0.2\,\mathrm{K}$, $\Delta p = 0.3\,\mathrm{hPa}$ and $\Delta H = 1.2\,\%$, following typical uncertainties suggested in the sensors' user manuals.

Other uncertainty contributions are set to $\Delta u_{\mathrm{li}} = 0.1\,\mathrm{m\,s}^{-1}$ and $\Delta z_0 = 0.05\,z_0$. The wind speed uncertainty at measuring height $\Delta u_{\mathrm{m}}$ is dependent on the line-of-sight wind speed uncertainty $\Delta u_{\mathrm{LOS}} = 0.1\,\mathrm{m\,s}^{-1}$, the azimuth uncertainty $\Delta \vartheta = 0.05°$ and the wind direction uncertainty $\Delta \chi = 1°$, following error propagation of Equation 6. $\Delta z_m$ was estimated using the pitch and roll uncertainty, which were set to $\Delta \beta = \Delta \gamma = 0.05°$ following the findings of the method of sea surface levelling demonstrated

in Rott et al. (2017, 2020). These uncertainties can be understood to comprise both possible elevation pointing uncertainties as well as the tilt of the lidar device. All uncertainty terms defined here and thus also the total propagated uncertainty $\Delta u_{\mathrm{TP}}$ are understood as the $1.96\,\sigma$ values of the corresponding error distributions, i.e. we expect them to include 95% of all values.

A detailed analysis of the uncertainty associated with wind speed extrapolation to hub height in the framework of an offshore lidar campaign by Theuer et al. (2020a) has revealed a strong dependency on the Obukhov length $L$. Large uncertainties need to be expected especially during very stable atmospheric conditions. Even though the study uses different input parameters, this also holds valid for our analysis.

We determined the total propagated uncertainty $\Delta u_{\mathrm{TP}}$ for each scan and grid point. Values were normalised within each scan $i$ using $\overline{u}_{\mathrm{TP},i}$ and averaged across all valid scans, yielding $\overline{\Delta u}_{\mathrm{TP,norm}}$.

### 2.5.2 Corrected propagated uncertainty

In the uncertainty estimation of the total propagated uncertainty (Section 2.5.1) we defined $\overline{\Delta u}_{\mathrm{TP,norm}}$ in a way that assumes none of the input uncertainties are correlated across range gates. That means, we also assume it is possible that the signs of the errors vary between range gates. While this might be true for wind speed errors $\Delta u_{\mathrm{m}}$ and roughness length errors $\Delta z_0$, it does not hold for measurement height errors $\Delta z_{\mathrm{m}}$, which are directly related to the tilt of the lidar, and the Obukhov length error $\Delta L$, which we consider to be constant across the whole measurement domain. Since these assumptions could influence the interpretation of the results, we decided to determine the uncertainty additionally only considering measurement range independent input variables. That means, we set $\Delta z_{\mathrm{m}} = \Delta \Psi_{\mathrm{TP}} = \Delta \Psi_{\mathrm{m}} = 0$ to calculate the corrected propagated uncertainty

$$\Delta u_{\mathrm{TP,cor}} = \left[ \left( \frac{\ln(\frac{z_{\mathrm{TP}}}{z_0}) - \Psi_{\mathrm{TP}}}{\ln(\frac{z_{\mathrm{m}}}{z_0}) - \Psi_{\mathrm{m}}} \Delta u_{\mathrm{m}} \right)^2 + \left( \frac{u_{\mathrm{m}}(\ln(\frac{z_{\mathrm{TP}}}{z_0}) - \Psi_{\mathrm{TP}} + \Psi_{\mathrm{m}} - \ln(\frac{z_{\mathrm{m}}}{z_0}))}{z_0(\ln(\frac{z_{\mathrm{m}}}{z_0}) - \Psi_{\mathrm{m}})^2} \Delta z_0 \right)^2 \right]^{1/2} . \tag{15}$$

Also $\Delta u_{\mathrm{TP,cor}}$ is normalised within each scan and subsequently averaged across all valid scans to $\overline{\Delta u}_{\mathrm{TP,cor}}$.

We examine the uncertainty contributions of $\Delta z_{\mathrm{m}}$, $\Delta \Psi_{\mathrm{TP}}$ and $\Delta \Psi_{\mathrm{m}}$ for relevant cases separately in a case distinction in Section 3.4.

### 2.5.3 Standard error of the mean

As an alternative to the total propagated uncertainty we calculated the statistical error, i.e. the standard error of the mean

$$\mathrm{SEM} = 1.96 \frac{\sigma_{u_{\mathrm{TP,norm}}}}{\sqrt{N_r}} \tag{16}$$

for each grid point, considering all valid scans $N_r$ with the standard deviation of the normalised wind speed at each grid point $\sigma_{u_{\mathrm{TP,norm}}}$. We included the factor 1.96 already in the definition of the variable to cover the 95 % confidence interval for normally distributed errors. The SEM estimates the deviation of the sample mean from the true mean (McKillup, 2005) and thus yields information regarding the statistical significance of the results. While the total propagated uncertainty regards the accuracy of single input variables, the statistical error quantifies the precision of the results from different scans. A higher number of scans typically reduces measurement noise from the statistical error, i.e. wind speed fluctuations around the mean.

### 2.5.4 Uncertainty due to local wind direction deviations

In addition to the total propagated uncertainty, the corrected propagated uncertainty and the SEM, which focus on the uncertainty in mean wind direction, we want to introduce another uncertainty in the measured wind fields that arises from the assumption of a homogeneous wind direction in the whole scanned area. As described in Section 2.4 we estimate the mean wind direction of a single lidar scan by means of a cosine fit (VAD algorithm) and transfer all measured line of sight wind speeds of the current scan to absolute horizontal wind speeds using this fixed mean wind direction (Equation 6). Local deviations from this mean wind direction lead to estimation errors of the horizontal wind. This uncertainty contribution is dependant on the angular difference between the wind direction and the lidar scanner's azimuth angle and the degree of the deviation. To quantify this we define the local uncertainty

$$\frac{u_{\mathrm{h}}}{u_{\mathrm{true}}} = \frac{\frac{u_{\mathrm{LOS}}}{\cos(\vartheta - \chi)}}{\frac{u_{\mathrm{LOS}}}{\cos(\vartheta - \chi_{\mathrm{true}})}} = \frac{\cos(\vartheta - \chi_{\mathrm{true}})}{\cos(\vartheta - \chi)} \tag{17}$$

with the local estimated wind speed $u_{\mathrm{h}}$ and the true local wind speed $u_{\mathrm{true}}$ that considers wind direction deviations being dependant on the lidar's azimuth angle $\vartheta$ and the local wind direction $\chi_{\mathrm{true}} = \chi + \chi_{\mathrm{div}}$ being the sum of the mean wind direction $\chi$ and the local wind direction deviation $\chi_{\mathrm{div}}$.

Figure 3 visualizes this error for a hypothetical lidar scan with a constant deviation of $|\chi_{\mathrm{div}}| = 2.0°$ and its sign chosen corresponding to a flow around the wind farm. While the error vanishes for the lidar looking upwind it leads to reduced wind speed estimates of more than 4 % at the sides of the scan ($\vartheta - \chi = \pm 50°$) in this example. Since this uncertainty contribution is in the same order of magnitude as the expected strength of global blockage at the sides of the scan we focus our analysis on the upstream direction where the error is neglectable.

## 3 Results

In the following we present results of the four scenarios introduced in Section 2.4.

### 3.1 Scenario 1: Wind farm operating below rated wind speed at unstable atmospheric conditions with high thrust coefficient

Figure 4 (a) shows $N = 53$ normalised and averaged lidar scans for unstable atmospheric conditions and within a wind speed interval of $\overline{u} = [8 - 11]\,\mathrm{m\,s^{-1}}$, i.e. for the wind farm operating at a high thrust coefficient. The wind field depicted is relatively homogeneous across the shown area, with slight variations of wind speed visible as streaks in wind direction. Apart from that, most values fluctuate closely around the mean wind speed.

This impression is confirmed by Figure 4 (b), where a virtual cut in mean wind direction upstream of the lidar, indicated as red line in Figure 4 (a), is depicted. Again, $\overline{u}_{\mathrm{TP,norm}}$ fluctuates around 1, the three error terms (c. f. Equations 9, 15 and 16), visualised as light and dark grey shaded areas and black dotted line respectively, have a similar magnitude. Be aware that the SEM overlays the corrected propagated uncertainty $\overline{\Delta u}_{\mathrm{TP,cor}}$ and makes its shaded area seem darker than depicted in the

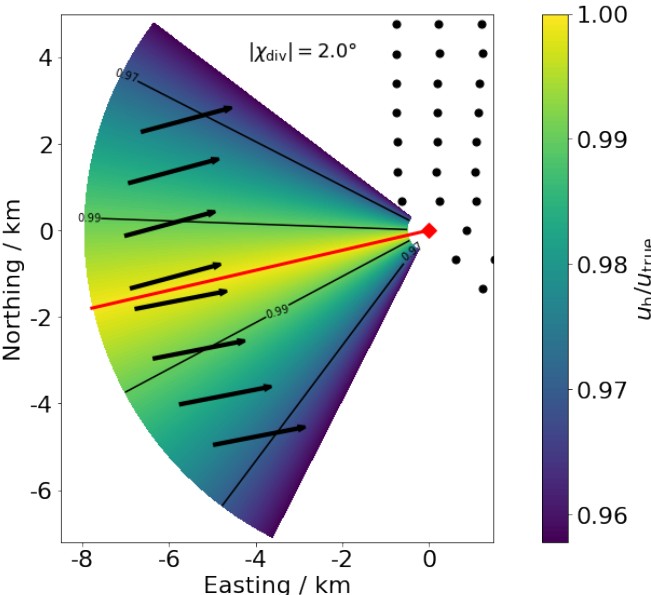

**Figure 3.** Local deviation in the assessment of the horizontal wind speed $u_{\mathrm{h}}$ due to a wind direction deviation $|\chi_{\mathrm{div}}|$, here exemplary set to $2.0°$ (cf. Equation 17). The lidar (red $\diamond$) directs its beam (red) in mean wind direction $\chi$, arrows denote the local deviation of the wind direction $\chi_{\mathrm{div}}$.

legend. The uncertainty contributions of $L$ and $z_{\mathrm{m}}$ are relatively low, i.e. $\overline{\Delta u}_{\mathrm{TP,norm}}$ is only slightly larger than $\overline{\Delta u}_{\mathrm{TP,cor}}$. This can be attributed to the relatively small change of wind speed with height during unstable conditions. Generally slightly larger values can be observed for the SEM as compared to $\overline{\Delta u}_{\mathrm{TP,cor}}$. For far range gates from approximately $-33D$ onward, the SEM increases significantly as a consequence of lower data quality and the lower number of values considered here (cf. Figure 4 (c)). We found no evidence for a decreasing trend in wind speed upstream of the wind farm GT I for Scenario 1.

**3.2    Scenario 2: Wind farm not operating at stable atmospheric conditions.**

Figure 5 shows normalised and averaged wind speeds in the inflow region of GT I for stable atmospheric conditions and wind speeds below cut-in in the same manner as Figure 4. Strong relative variations of $\overline{u}_{\mathrm{TP,norm}}$ from the average wind speed are visible across the scan. Larger values occur for low azimuth angles, i.e. south of the wind farm. Two wakes, located to both sides of the wind field-cut, with diminished wind speed are visible. These wakes are likely caused by jack-up barges used to 360    construct the two neighbouring wind farms. Despite the large wind speed fluctuations within the scan area, no decrease of wind speed with decreasing distance to the wind farm can be observed, neither in Figure 5 (a) nor (b). The measurements show a slight increase in the wind speed of approximately 3% up to approximately $-30D$ and decrease again closer to the wind farm. One should be aware that due to the low average wind speed ranging from $\overline{u} = 3 - 4\,\mathrm{m\,s^{-1}}$ with a mean value of $\overline{u}_{\mathrm{TP}} = 3.48\,\mathrm{m\,s^{-1}}$ the large percental variation in wind speed across the scan is low in absolute values.

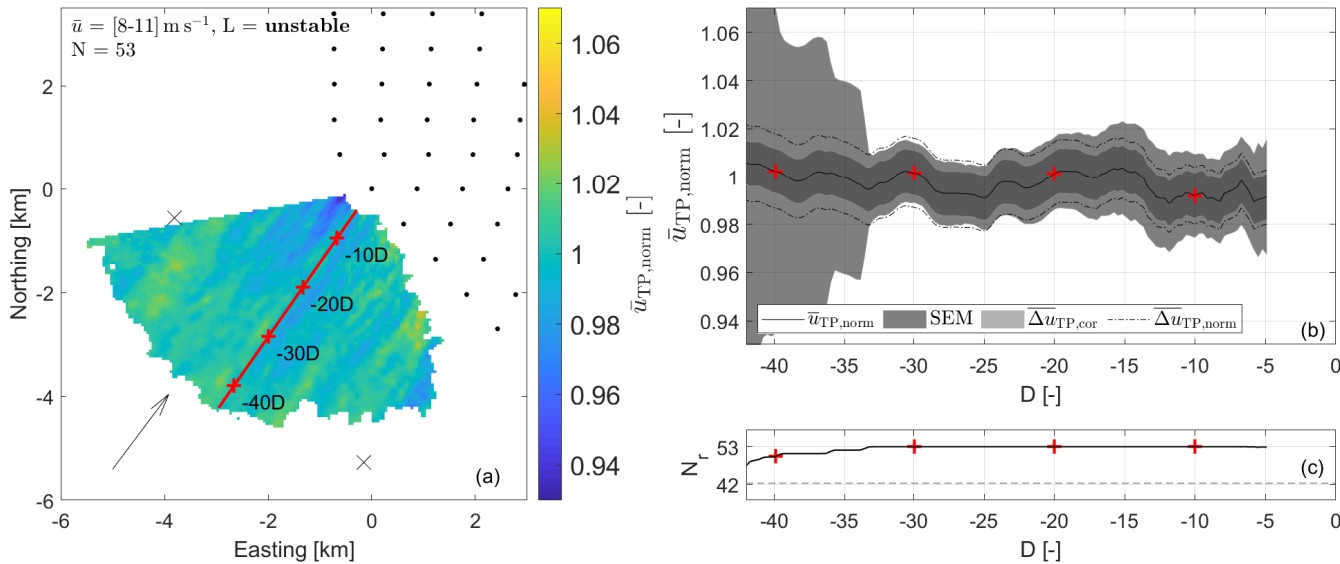

**Figure 4.** Scenario 1: Analysis of lidar scans during unstable atmospheric conditions and in cases with high thrust coefficient with $\overline{u} = [8 - 11]\,\mathrm{m\,s^{-1}}$, $\chi = [210 - 220]°$, $N = 53$, $\overline{\chi} = 217°$, $\overline{u}_{\mathrm{GT58}} = 10.11\,\mathrm{m\,s^{-1}}$, $\overline{P}_{\mathrm{GT58}} = 4.77\,\mathrm{MW}$, the median Obukhov length $L_{\mathrm{med}} = -430\,\mathrm{m}$ and the standard deviation of the wind direction $\sigma_\chi = 2.68°$. Subfigure (a) depicts the normalised wind speed $\overline{u}_{\mathrm{TP,norm}}$ averaged over all valid scans $N$. The arrow displays the mean wind direction $\overline{\chi}$. $\overline{u}_{\mathrm{TP,norm}}$ along the wind field-cut, indicated as red line in (a), is shown in Subfigure (b). Here, additionally the three estimated uncertainties $\overline{\Delta u}_{\mathrm{TP,cor}}$, SEM and $\overline{\Delta u}_{\mathrm{TP,norm}}$ are visualised as grey shaded areas and black dotted line respectively. Be aware that the SEM here overlays $\overline{\Delta u}_{\mathrm{TP,cor}}$. The distance to the lidar on the x-axis is given in terms of rotor diameter $D$. Subfigure (c) displays the number of valid scans at each grid point $N_r$. The grey horizontal dashed line marks $0.8N$. Points highlighted by red x's in (b) and (c) correspond to the locations marked in the lidar-cut in (a) and (b).

Especially distinct are the large values of $\overline{\Delta u}_{\mathrm{TP,norm}}$ shown in (b). We attribute this to the very stable atmospheric conditions, which cause large wind speed extrapolation uncertainties (Theuer et al., 2020a) and low wind speeds, which result in larger relative uncertainties. Be aware of the different y-scale as compared to Figures 4, 6 and 7. The SEM is of similar size as $\overline{\Delta u}_{\mathrm{TP,cor}}$, however, it strongly increases with decreasing $N_r$ for increasing distances to the wind farm (Figure 5 (c)), exceeding $\overline{\Delta u}_{\mathrm{TP,cor}}$ and finally $\overline{\Delta u}_{\mathrm{TP,norm}}$ for distances larger than $-40D$.

All shown uncertainty ranges are able to account for the observed variations in wind speed. Taking into account also its aforementioned low absolute values, we consider these variations to be insignificant.

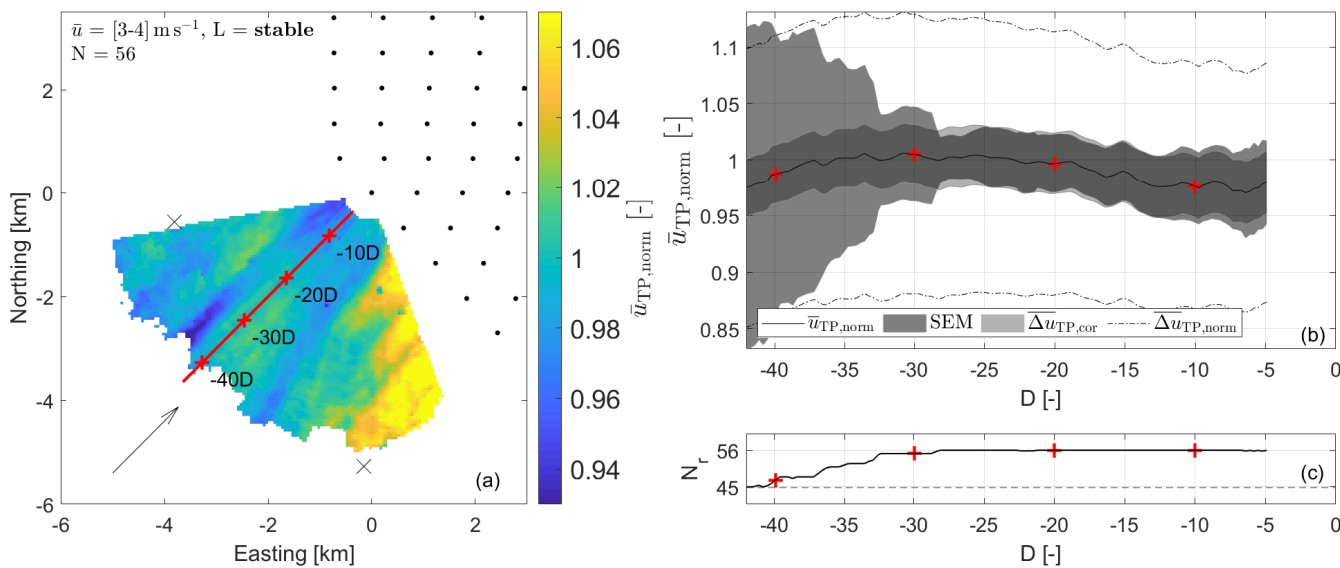

**Figure 5.** Scenario 2: Analysis of lidar scans during stable atmospheric conditions and in cases with low thrust coefficient (turbines not operating) with $\overline{u} = [3-4] \, \mathrm{m \, s^{-1}}$, $\chi = [220-230]°$, $N = 56$, $\overline{\chi} = 225°$, $\overline{u}_{\mathrm{GT58}} = 3.48 \, \mathrm{m \, s^{-1}}$, $\overline{P}_{\mathrm{GT58}} = 0.06 \, \mathrm{MW}$, $L_{\mathrm{med}} = 54 \, \mathrm{m}$ and $\sigma_{\chi} = 2.84°$. For details on Subfigure (a), (b) and (c) refer to the caption of Figure 4. Be aware that the SEM here overlays $\overline{\Delta u}_{\mathrm{TP,cor}}$.

### 3.3 Scenario 3: Wind farm operating above rated wind speed at stable atmospheric conditions with low thrust coefficient

Figure 6 visualises the results of Scenario 3, considering stable atmospheric conditions and the wind farm running far beyond
rated wind speed, i.e. at a low thrust coefficient of $\leq 0.3$. No decrease of wind speed close to the wind farm as indication of global blockage can be observed with $\overline{u}_{\mathrm{TP,norm}}$ fluctuating around a value of 1. As shown in Figure 6 (b), similarly as for Scenario 1, the SEM exceeds $\overline{\Delta u}_{\mathrm{TP,cor}}$ and increases strongly for far distances due to decreasing $N_r$ (see Figure 6 (c)) and decreasing data quality for far range gates. The difference between $\overline{\Delta u}_{\mathrm{TP,cor}}$ and $\overline{\Delta u}_{\mathrm{TP,norm}}$ increases with distance to the wind farm as a consequence of the increasing measuring altitudes and the shape of the wind profile. Also considering the larger
mean wind speed, the impact of the uncertainties of $L$ and $z_{\mathrm{m}}$ is here stronger as compared to the unstable cases in Scenario 1.

### 3.4 Scenario 4: Wind farm operating below rated wind speed at stable atmospheric conditions with high thrust coefficient

In Figure 7 we illustrate the results of Scenario 4, i.e. stable atmospheric conditions and the wind farm operating at a high thrust coefficient. The normalised and averaged wind field shown in Figure 7 (a) suggests a decrease in wind speed for flow
approaching the wind farm. Contour lines highlight the shape of the decrease and show that it is less distinct at the sides of

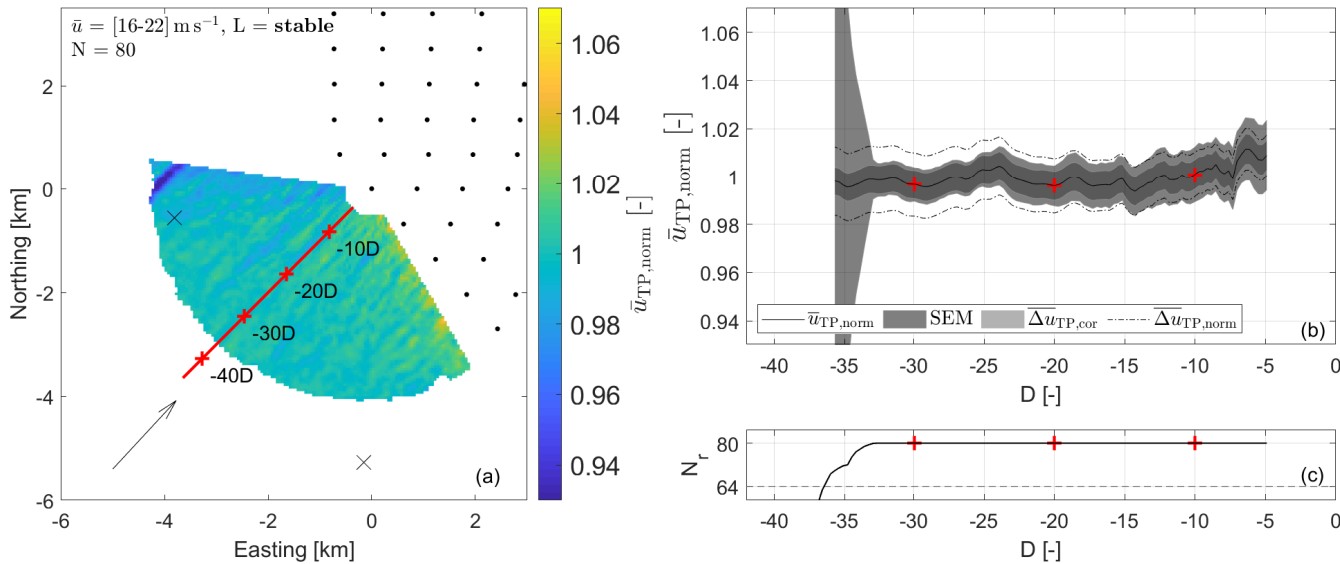

**Figure 6.** Scenario 3: Analysis of lidar scans during stable atmospheric conditions and in cases with low thrust coefficient with $\overline{u} = [16 - 22]\,\mathrm{m\,s^{-1}}$, $\chi = [220 - 230]^\circ$, $N = 80$, $\overline{\chi} = 223^\circ$, $\overline{u}_{\mathrm{TP}} = 19.83\,\mathrm{m\,s^{-1}}$, $\overline{P}_{\mathrm{GT58}} = 4.79\,\mathrm{MW}$, $L_{\mathrm{med}} = 460\,\mathrm{m}$ and $\sigma_\chi = 1.54^\circ$. For details on Subfigure (a), (b) and (c) refer to the caption of Figure 4. Be aware that the SEM here overlays $\overline{\Delta u}_{\mathrm{TP,cor}}$.

the lidar scan. Observed values of $\overline{u}_{\mathrm{TP,norm}}$ vary between approximately 0.96 and 1.04. The virtual cut on the wind field given in Figure 7 (b) supports these findings. Starting at a value of about 1.03 at $-44\,D$ the normalised and averaged wind speed slowly decreases until it reaches a value of 0.99 at $-5\,D$. The magnitude of the curve's slope hereby increases with decreasing distance to the wind farm. The SEM is narrow compared to $\overline{\Delta u}_{\mathrm{TP,cor}}$ and especially $\overline{\Delta u}_{\mathrm{TP,norm}}$. Only for far

distances it strongly increases as a consequence of reduced sample size (see Figure 7 (c)) and data quality. The total propagated uncertainty $\overline{\Delta u}_{\mathrm{TP,norm}}$ reaches values of up to 3 %. As the analysed scans are attributed to stable atmospheric conditions, the impact of $L$ and also $z_{\mathrm{m}}$ is large.

As defined in Section 2.5, the width of the uncertainty contributions are considered to cover 95 % of all cases. Figure 7 (b) indicates a significant decrease of wind speed closer to the wind farm when considering the corrected propagated uncertainty

$\overline{\Delta u}_{\mathrm{TP,cor}}$. This is not true anymore when including all error contributions, i.e. considering the width of the total propagated uncertainty $\overline{\Delta u}_{\mathrm{TP,norm}}$. In Figure 8 we visualise how the decrease of wind speed changes when assuming the largest uncertainties for $L$ and $z_{\mathrm{m}}$ to analyse the error contributions of the range gate-correlated variables in more detail. We consider the same data set in Figure 8 as in Figure 7. Here, we show the two most extreme scenarios with $L - \Delta L, z_{\mathrm{m}} + \Delta z_{\mathrm{m}}$ (blue, largest reducing effect on the deficit) and $L + \Delta L, z_{\mathrm{m}} - \Delta z_{\mathrm{m}}$ (red, largest enhancing effect on the deficit) respectively. As explained

earlier, we assume $\Delta L$ and $\Delta z_{\mathrm{m}}$ to be correlated across range gates within the same scan and thus consider the corrected propagated uncertainty $\overline{\Delta u}_{\mathrm{TP,cor}}$ more valuable than the total propagated uncertainty $\overline{\Delta u}_{\mathrm{TP,norm}}$ depicted in Figure 7 (b).

As clearly visible in Figure 7 (b) and Figure 8, misestimations of Obukhov length and measurement height have a significant impact on the magnitude and shape of the observed wind speed decrease. In the blue graph in Figure 8 the wind speed deficit is reduced as a consequence of the more stable conditions and larger differences between measuring height and hub height assumed here. Considering the associated uncertainty, the observed wind speed deficit of approximately 2 % for this case with the largest reducing effect tends to be within the range of the corrected propagated uncertainty. When considering errors with less reducing effect, i.e. errors with the same sign but smaller magnitude, the wind speed deficit increases towards a significant value. If maximal errors occur in the opposite direction (red curve), the effect would be maximally enhanced to a wind speed decrease of 6 %. Here, the observed decrease is large compared to the uncertainty intervals and thus clearly significant. Considering the range gate-correlated error contributions the wind speed deficit of 4 % lies within an uncertainty interval between 2 % and 6 %.

Although we only show one wind direction sector, we observed similar wind speed deficits for different sectors. Here we show the most distinct case.

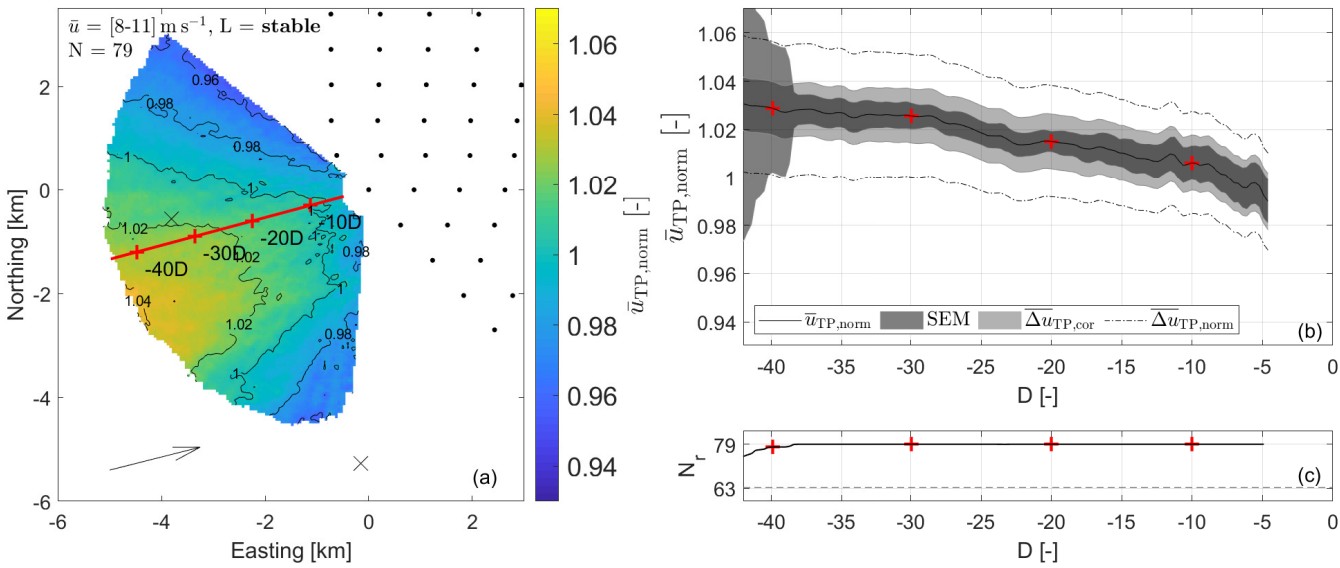

**Figure 7.** Scenario 4: Analysis of lidar scans during stable atmospheric conditions and in cases with high thrust coefficient with $\overline{u} = [8 - 11]\,\mathrm{m\,s^{-1}}$, $\chi = [250 - 260]°$, $N = 79$, $\overline{\chi} = 256°$, $\overline{u}_{\mathrm{GT58}} = 9.23\,\mathrm{m\,s^{-1}}$, $\overline{P}_{\mathrm{GT58}} = 4.51\,\mathrm{MW}$, $L_{\mathrm{med}} = 307\,\mathrm{m}$ and $\sigma_\chi = 2.78°$. Contour lines in the flow field in Subfigure (a) highlight the shape of the wind field upstream GT I. For further details on Subfigure (a), (b) and (c) refer to the caption of Figure 4.

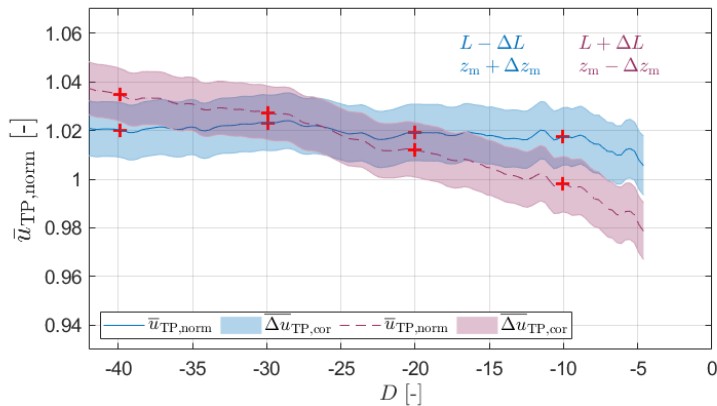

**Figure 8.** Scenario 4: $\overline{u}_{\mathrm{TP,norm}}$ over the distance to the lidar given in terms of rotor diameter $D$ along the wind field-cut, indicated as red line in Figure 7 (a), with the corrected propagated uncertainty $\overline{\Delta u}_{\mathrm{TP,cor}}$ visualised as colored shaded area. For the blue curve $L$ was additionally reduced by $\Delta L$ and $z_{\mathrm{m}}$ increased by $\Delta z_{\mathrm{m}}$, for the red curve $L$ was increased and $z_{\mathrm{m}}$ reduced respectively. These cases represent the two combinations of $\Delta L$ and $\Delta z_{\mathrm{m}}$ that yield the most extreme results.

## 4 Discussion

We analyzed averaged long-range Doppler lidar PPI scans at TP height in the inflow of the 400 MW offshore wind farm GT I and found wind speed deficits upstream in stably stratified boundary layers with wind turbines operating at high thrust coefficient in the upper partial load range. In contrast, at unstable stratification and similar operating conditions, no effect was visible. We identified the comparably small wind speed difference by performing a data correction and by averaging the normalised lidar scans. We analyzed the effect considering a detailed uncertainty estimation. In this section, we discuss our findings and relate them to the global blockage effect dependant on atmospheric stability and to the wind farm's operational state as well as possibilities and difficulties for global blockage measurements.

### 4.1 Global blockage dependant on atmospheric and operational conditions

To distinguish between different wind turbine operational states and atmospheric stabilities we divided our measurement data into four different scenarios (cf. Section 2.4).

**Scenario 1:** In unstable conditions with wind speeds from $8\,\mathrm{m\,s}^{-1}$ to $11\,\mathrm{m\,s}^{-1}$ and a high thrust coefficient (Scenario 1, Figure 4) we could not identify decreasing wind speeds in front of the wind farm and thus no global blockage effect. This result is plausible since wind speed fluctuations in unstable flows are much higher due to convection than the assumed magnitude of global blockage. Convection leads to more mixing in the boundary layer and thus repeals global blockage due to vertical transport of momentum. Furthermore, in unstable stratification, the boundary layer is typically higher and thus the flow can pass obstacles like hills (Stull, 1988) or in this case a wind farm more easily. The effect of atmospheric stability could be investigated further using high fidelity simulation like LES in future studies.

Additionally to Scenario 1 we performed the analysis for unstable stratification and the wind speed ranges above rated wind speed and below cut-in wind speed respectively (cf. Section 2.4). In both cases, we could not identify decreasing wind speeds in the inflow of the wind farm. As explained earlier we do not show these results here for brevity.

**Scenarios 2 and 3:** In stable atmospheric stratification and with low wind turbine thrust coefficients due to low wind speeds (i.e. not operating turbines, Scenario 2, cf. Figure 5) or due to high wind speeds (i.e. turbines operating with pitched blades above rated wind speed, Scenario 3, cf. Figure 6) no wind speed reductions upstream of the wind farm were identifiable. When the turbines are out of operation there should not be any reason for global blockage to appear due to the very low thrust. For turbines operating above rated wind speed a small global blockage effect might occur. However, it is unlikely that the effect

would be clearly visible in the data as a consequence of high wind speeds and the reduced thrust. Since the turbines operate at rated power global blockage, if any, would not have a negative impact on power production in this wind speed range.

**Scenario 4**: In stable atmospheric conditions with a high wind turbine thrust coefficient (i.e. wind turbines operating in the partial load range, Scenario 4, cf. Figure 7) we found the wind speed to decrease towards the wind farm by approx. 4 % over a distance of 25 $D = 2.9$ km. For larger distances upstream, the wind speed approaches an almost constant value with no further

increase visible. We assume the wind speed at 40 $D$ upstream to be free stream speed but can not be sure whether small wind farm induced effects reach even further. The wind speed reduction is significant when considering range gate-uncorrelated uncertainties. It is considered meaningful for global blockage to be most significant in stable stratification and for higher thrust coefficients.

**Is the observed wind speed deficit global blockage?** Despite the intensive uncertainty analysis and error correction we per-

450 formed in this work (cf. Section 2.5), how certain can we be that our observations are caused by the global blockage effect? To consider the effect of the correlated error sources that we excluded from the calculation of the total propagated uncertainty, namely $\Delta L$ and $\Delta z_{\mathrm{m}}$, Figure 8 shows the two most extreme cases with the largest reducing effect (blue) and the largest enhancing effect (red) of both error values. Assigning these combinations of errors, the extent of the global blockage wind speed deficit is limited by 2 % to 6 %. In the latter case, the wind speed deficit is clearly significant, while for the first one, it could be

explained by the correlated propagated uncertainty. However, considering more likely error magnitudes in between those two most extreme cases, the wind speed deficit would become significant. Thus, we consider the wind speed deficit in front of the wind farm to be caused by the global blockage effect.

**Spatial inhomogenities in the wind field:** Different from wind tunnel measurements where all background conditions could be controlled, free field measurements cover numerous superimposing meteorological effects. In our case especially the back-

460 ground wind field without the presence of wind farms needs consideration. The small flow effects we observe here are in the order of magnitude as typical wind field fluctuations locally (background turbulence) and over distances of some kilometres (spatial wind field variations). In a single lidar scan it would be not clear whether a wind speed gradient in front of a wind farm is caused by the global blockage effect or by a spatial variation in the background wind field. Our method using the average wind field of lots of lidar scans from different days allows us to average different spatial wind speed gradients of the

465 background flow. In Scenarios 1-3 we found spatial wind speed gradients that are almost zero in average. Scenario 4 shows a negative wind speed gradient. We assume this to be caused mainly by global blockage with the background turbulence and the

spatial variation averaged out.

**The effect of measurement height:** Our measurements of global blockage were performed at a height of approx. 9 m below the rotor area while Bleeg et al. (2018) used mainly measurements at hub height and some at 70 % of the hub height. An extrapolation to hub height instead of lidar height would not have a significant impact on our findings as it would only result in an upscaling of the observed effect to a higher altitude. Further, we assume extrapolation uncertainties would increase significantly when extrapolating across larger height differences (Theuer et al., 2020a). We do not know the vertical distribution of the global blockage effect but expect it to be equally distributed between the surface and upper blade tip height. Bleeg et al. (2018) found only small variations in the analyzed blockage effect comparing mast measurements at hub height and 70 % of the hub height. CFD results indicate a rather constant global blockage effect up to at least hub height due to the presence of the ground (Branlard et al., 2020). In this investigation, we did not study global blockage induced flow deflections upwards, downwards or sideways which could lead to increased wind speeds above, below or aside the wind farm's rotor area. The vertical extent of the global blockage effect in front of a wind farm needs to be assessed in future experimental studies to verify numerical results.

**Influences of cluster wakes:** Schneemann et al. (2020) show the existence of cluster wakes in the inflow of GT I using data from the same measurement campaign as used here. In the wind directions we chose for the analysis of global blockage, no distinct wind speed gradients are present in the inflow. Schneemann et al. (2020) did not find signatures of the wakes of single wind turbines in the inflow of GT I. Nygaard and Newcombe (2018) showed dual Doppler radar measurements of a wind farm wake with the signatures of single wind turbines disappearing less than 10 km behind the farm. Generally, the data we present here could be influenced by cluster wakes. However, we do not expect disturbances of the global blockage measurement since the centre flow of a cluster wake in the far-field is comparably homogeneous and would only reduce the mean wind speed in the whole lidar scan. Recovery of the possible cluster wake influence with a positive wind speed gradient towards GT I could on the other hand contradict the global blockage effect with a negative wind speed gradient. This would lead to a reduction of the observable global blockage effect. Furthermore, cluster wakes could have an influence on the prevailing wind profile that can not be quantified here. This is subject of current research.

**The magnitude of global blockage:** We found the magnitude of the global blockage induced deficit of approx. 4 % (uncertainty range 2 % to 6 %) in stable stratification to correspond well with values measured in an onshore free field experiment by Bleeg et al. (2018) based on met mast point measurements at three different wind farms of typically 2 % to 4 %. One possible explanation of the comparably lower deficits of Bleeg et al. (2018) is the lack of stability information and thus the comparison of long intervals including the climate mean of stratifications. Our results suggest less or no global blockage effects in unstable stratification, this effect possibly reduced the average values of Bleeg et al. (2018). RANS simulations performed by Bleeg et al. (2018) typically showed similar or slightly smaller global blockage deficits. Wu and Porté-Agel (2017) found global blockage deficits in LES simulations of large finite-size wind farms of 1.3 % and 3 % for different farm layouts in a weak free-atmosphere with neutral stratification across the rotor area, which is slightly lower than our findings. When discussing the strength of global blockage in our data, we need to consider the measurement distances. We analyzed a wind speed difference between 40 D and 4 D upstream. On the far distance the effect seems to have almost vanished with a constant wind speed.

Nevertheless, a further slight increase in wind speed for larger distances is possible. Moreover, the strong wind speed gradients at the lower distance of 4 D suggest an even further wind speed decrease towards the wind farm. Therefore, we assume the global blockage effect to be even stronger than quantified here.

**The spatial extent of global blockage:** Aside results from wind tunnel experiments (Segalini and Dahlberg, 2019) and onshore free field point measurements (Bleeg et al., 2018) our lidar measurements represent the first areal free field investigation of the global blockage effect offshore. Spatial analysis of global blockage has only been reported from numerical studies so far. RANS simulations performed by Bleeg et al. (2018) for three different large onshore wind farms reveal homogeneous induction zones upstream of the farms with spatial extents of more than 2 km for a deficit of 1 %. Such distances correspond

well to our findings in Scenario 4. The higher wind speed deficits in our data could possibly be explained by the restriction to stable stratification. Nevertheless, the shape of the induction zones in their RANS simulations seem to smoothly follow the first row of turbines. The contours we show in Figure 7 tend to have the same shape in the middle sector of the wind field but deviate from that shape on the sides. We assume this behaviour to be mainly related to the applied assumption of a homogeneous wind direction. With an increasing angular difference of the lidar's azimuth angle from the wind direction the

orthogonal components of the main wind direction, i.e. local deviations in the wind direction increasingly contaminate the measurement of the local horizontal wind under the assumption of a homogeneous wind direction. (cf. Section 2.5.4). Flow effects resulting from the wind farm's blockage with wind components tending to flow around the wind farm (Porté-Agel et al., 2019) can influence the observed contours. The exemplary local deviation of the estimated horizontal wind field we show in Figure 3 assumes a diverging flow with wind direction deviations of $2.0°$. This low value of divergence could well explain

the observed shape of the contours with underestimated horizontal wind speeds of more than 4 %. Future experimental studies should focus on assessing the global blockage induced flow around the farm from dual Doppler lidar measurements or single lidar measurements with more advanced analysis techniques.

**The influence of global blockage on power production:** To assess the impact of the global blockage effect on a wind farms annual energy production (AEP) more research and development on the implementation and validation of the effect in wind

farm planning tools is needed. A detailed AEP assessment then needs to consider particularly the local undisturbed wind speed and stability wind roses. The consideration of global blockage in the future could further increase the accuracy of wind energy site assessment which is especially important for the financing process of wind farm projects. Despite its possible negative impact on energy production, global blockage seems not to have a critical impact on wind energy utilization. In our study, we observed the effect only below rated power, in stable stratification and with a magnitude of 4 % within the uncertainty range of

2 % to 6 %. Consequently, we expect global blockage to have a much lower impact on the power production than other wind farm flow features like inner wind farm wakes.

## 4.2    Global blockage measurement techniques

Compared to wind turbine wake effects with several $\mathrm{ms}^{-1}$ wind speed deficit over a distance of less than one rotor radius between wake centre and free flow the global blockage effect has a comparably small magnitude. Scenario 4 reveals a deficit

of about 4 % of the average wind speed of $9.2\,\mathrm{ms}^{-1}$ which equals approx. $0.4\,\mathrm{ms}^{-1}$. This difference builds up over a distance

of 25 $D = 2.9$ km. This is well below typical fluctuations in wind fields due to e.g. orographic or thermal influences which makes global blockage hard to identify in measurement data.

There are no further areal wind field measurements of global blockage in literature. The shape of the zone with reduced wind speeds in front of the wind farm and comparisons of different locations could not be analyzed using single-point measurements like presented by Bleeg et al. (2018). Spatial characteristics of global blockage inflows of wind farms that were generated by numerical simulations and modelling (Bleeg et al., 2018; Branlard and Meyer Forsting, 2020) have not been experimentally verified, yet. Different from point measurements areal lidar wind field measurements of the wind farm inflow resolving the zone with wind speed reduction can allow for a more detailed analysis.

Generally, we do not expect global blockage to be significantly identifiable in single flow measurements like an individual lidar scan. The effect is much smaller than the common fluctuations in wind farm inflows and needs to be derived from averaged measurements where the influence of local turbulence and coherent turbulent structures is reduced in the averaging process.

The lidar measurements we analyse in this paper were originally performed to study the effect of cluster wakes in the inflow of GT I (Schneemann et al., 2020) and to perform minute-scale power forecasts (Theuer et al., 2020b). Due to the comparably small global blockage effect, all errors influencing the accuracy of lidar measurements need to be carefully examined and reduced wherever possible. We give an overview on sources of uncertainty in Table 2. For lidar measurement campaigns aiming at the assessment of global blockage or similar small flow effects we recommend to

- calibrate the lidar before the campaign. This includes the measurement of radial velocities, the range gate distance from the device and especially the scanner orientation and movements. Here especially the scanners elevation angle deviation is crucial since it results in height errors of the measurement.

- carefully align the lidar at the measurement location and to monitor the lidar's tilt dynamically. We recommend using accurate inclinometers and accelerometers and in offshore campaigns the method of "sea surface levelling" for lidar tilt alignment and the method of "hard targeting" for alignment of the north direction (Rott et al., 2017, 2020).

- perform independent measurements of the prevailing wind profile either by e.g. met mast, VAD lidar or virtual met masts spanned by scanning lidars (Bell et al., 2020) to be able to perform a proper height correction of the scanning lidar data.

- perform measurements of meteorological quantities for characterization of atmospheric stability to support a more precise interpolation of the wind profile (e.g. Schneemann et al., 2020).

The stronger tilting on the nacelle compared to the transition piece and the resulting large errors in the measurement height introduce increased uncertainties to nacelle-based measurements especially when aiming to achieve several kilometres of range or to detect small flow effects like global blockage. Active motion compensation of the lidar's scanner or similar measures could enable the possibility of nacelle-based measurements.

Further, the use of overlapping Dual Doppler measurements could be beneficial to better resolve local flow characteristics like global blockage induced flow deflections and to overcome the need for basic assumptions like the homogeneity of the wind field in the VAD algorithm (e.g. van Dooren et al., 2016; Stawiarski et al., 2013). Another measurement system to assess global

blockage could be the remote sensing method Doppler radar which was successfully deployed for wind turbine and wind farm wake measurements (Nygaard and Newcombe, 2018).

## 5  Conclusions

This paper has pursued the objective to analyze whether it is possible to measure global wind farm blockage with long-range Doppler lidar dependant on different atmospheric stability estimates and wind farm operational states. We present averaged lidar PPI measurements of the inflow of the 400 MW offshore wind farm Global Tech I. In stable stratification and with the turbines operating below and up to rated power with a high thrust coefficient, the measurements revealed reduced wind speeds at the height of the transition piece in the approaching flow. At unstable stratification and similar operating conditions, however, no effect was visible. We relate this upstream wind speed reduction to the presence of the wind farm, namely to global wind farm blockage. Therefore, we conclude global blockage to be dependant on atmospheric stability.

Compared to wind turbine wakes or cluster wakes, global blockage is a very small effect that is overlaid with different atmospheric phenomena and thus very hard to detect. Nevertheless, based on our detailed uncertainty assessment we arrive at the conclusion that the wind speed deficit in front of Global Tech I in our lidar measurements is caused by global blockage. Generally, we assume long-range Doppler lidar to be able to accurately measure global blockage and recommend to carefully align and calibrate the used lidar systems.

Our measurements agree with recent findings of the magnitude of the global blockage effect to range from 2 % to 6 %. At platform level, we found a wind speed reduction of 4 % within an uncertainty range from 2 % to 6 %, over a distance of approx. 2.9 km or 25 $D$. The influence of the global blockage effect on the annual energy production of a wind farm requires further experimental and numerical investigations. Due to the expected limited appearance of global blockage only in special atmospheric situations and wind farm operational states and its small magnitude, we expect the impact on power production to be much smaller in comparison to inner wind farm wakes. Accurate estimates of the global blockage effect by means of well-calibrated engineering models could further decrease uncertainties in wind farm site assessments and power calculations in the future.

In this work, we demonstrated scanning long-range Doppler lidar to be a suitable tool to study global wind farm blockage and provide strong evidence for the existence of the global blockage effect for a wind farm with the turbines operating at high thrust coefficients in a stably stratified atmosphere.

*Data availability.* Lidar and meteorological data are not published and could be made available on request. The OSTIA data set can be obtained from http://marine.copernicus.eu. A set of GT I wind turbine coordinates can be found in https://www.openstreetmap.org.

*Author contributions.* JS initiated the research, performed the measurement campaign, was heavily involved in funding acquisition and research discussion, wrote Sections 1, 2.1, 2.2, 2.5.4, 4 and 5 and prepared Figures 1, 2 and 3. FT performed the data analysis, was heavily involved in research discussion, wrote Sections 2.3, 2.4, 2.5 and 3 and made Figures 4, 5, 6, 7 and 8. AR was heavily involved in research discussion and supported paper writing. MD initiated the research and was heavily involved in funding acquisition and research discussion. MK was heavily involved in funding acquisition and supervised the research. All authors contributed intensively to an internal review.

*Competing interests.* The authors declare no conflict of interest.

*Acknowledgements.* We performed the lidar measurements and parts of the work in the framework of the research projects «OWP Control» (FKZ 0324131A) and «X-Wakes» (FKZ 03EE3008) both funded by the German Federal Ministry for Economic Affairs and Energy on the basis of a decision by the German Bundestag. Frauke Theuer is supported by the German Federal Environmental Foundation (DBU) (Grant Nr. 20018/582). We acknowledge the wind farm operator Global Tech I Offshore Wind GmbH for providing SCADA data as well as their support of the work and the measurement campaign. Thanks to Met Office for making the OSTIA data set available. Special thanks to Stephan Voß, ForWind Oldenburg, for his great support in the conduction of the measurement campaign and to Julia Gottschall, Fraunhofer IWES, for an interesting discussion and helpful comments.

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

**Table 2.** Possible errors and uncertainties that might be introduced during the lidar measurement campaign and data analysis. In addition to a description of the uncertainty components, the measures we take to minimise those are stated and it is indicated whether they are considered in the uncertainty estimation.

| uncertainty components | Description | Measures |
|---|---|---|
| Azimuth / elevation pointing error | Internal unknown offsets of lidar scanner | Not corrected for, calibration prior to measurement campaign recommended. Considered in uncertainty estimation. |
| Movement / tilt of lidar | Uncertainties in pointing accuracy, in particular caused by a wind turbine thrust-dependent tilt of the device. Influences the measurement height of the device and varies with the range gate. | Empirical correction function for thrust dependent platform movement (Rott et al., 2020). Considered in uncertainty estimation. |
| Curvature of the Earth | Systematic variation in measuring height | Corrected for, not considered in uncertainty estimation |
| Tide | Uncertainty in measuring height estimated to be $\pm 0.6\,\text{m}$ here | Not corrected for, not considered in uncertainty estimation |
| Uncertainty in elevation angle due to earth's curvature, lidar tilt and scanner pointing error | Transfer of tilted LOS wind speed to horizontal wind speed $(1/\cos)$ | Here less then $1°$ in total, contribution neglectable. Not corrected for, not considered in uncertainty estimation. |
| LOS wind speed | Uncertainty in LOS wind speed causes uncertainty in horizontal wind speed | Not corrected for, considered in uncertainty estimation |
| Assumption of a homogeneous wind field | Wind speed error as a consequence of wind field reconstruction (VAD algorithm) | Not corrected for, not considered in uncertainty estimation |
| Uncertainties in meteorological measurements | Results in uncertainties in stability estimation | Not corrected for, considered in uncertainty estimation |
| Uncertainties in roughness length estimation | Results in uncertainties in wind profile estimation | Not corrected for, considered in uncertainty estimation |
| Inapplicability of the logarithmic wind profile | Occurs especially during stable atmospheric conditions (Theuer et al., 2020a; Peña et al., 2008), which might be related to the occurrence of e. g. kinks or low level jets (Møller et al., 2019). Leads to uncertainty in wind speed correction to lidar height. | Not corrected for, not considered in uncertainty estimation |
| Laser beam deflection due to thermal gradients in the lower boundary layer | Results in measurement height error | Not corrected for, not considered in uncertainty estimation. |