# Peer review of "Offshore wind farm global blockage measured with scanning lidar"

_Wind Energy Science, 2020_

## Referee Comment (RC1) · Emmanuel Branlard (Referee) · 21 Dec 2020

This is a thorough paper which presents measurement data of the global blockage effect using a scanning lidar. The authors have carefully accounted for uncertainties in the measurements, and found that blockage effects were mostly found in stable atmospheric configurations.

I have some general and specific comments, I hope that addressing some of them can improve the paper. Here are my general comments:

- The authors mention low and high thrust coefficients throughout the paper without defining the range that have been used.

- Some additional uncertainties may be worth mentioning:

[Figure]

- The logarithmic law assumes an "undisturbed" atmosphere, without a wind farm. The induction may affect the profile and hence the extrapolations performed in this study. I wonder if this can have an important impact and might be a bit of a chicken and egg issue since the profile use to extrapolate might depend on the induction effect.

- Uncertainty in the Ct curve (using the NREL5MW instead of manufacturer's curve)

- The number of samples used for averaging (but it is roughly the same for all scenarios, so that shouldn't affect your conclusions)

- It was not clear to me how the thrust coefficient was computed (which wind speed is used as a reference), and I wonder if this can have an effect on the categorization of the cases. It might make sense to use the average Ct over a set of turbines close to the lidar to make sure this Ct is representative of the farm (though this introduce further issues for the determination of reference wind speed for waked turbine..).

- There is likely a relationship between the wind direction, the wind farm layout, and the blockage effect. Scenario 4 has a fairly different wind direction than the others. To be fair, given the layout, it could be expected that the blockage effect would be smaller for this wind direction. It might still be worth mentioning/investigating. Also, the wind direction fluctuations within a measurement period might affect the averaging of the flow field, and potentially reduce the observed blockage effect. Maybe the variability of the wind direction could be reported?

I believe you make a strong point that the blockage effect is mostly seen in stable conditions, but I hope that addressing some of these comments will further lift any doubts. Again, congratulation for the work. I'll be looking forward to a revised version of the paper.

Emmanuel Branlard

Here are my specific comments:

l.52: I leaves this up to you, but since most of your references are fairly recent, I

wonder if you'd consider replacing the one you have on that line by the older version of this paper from 2015: "E. Branlard, A. R. Meyer Forsting, Using a cylindrical vortex model to assess the induction zone in front of aligned and yawed rotors, Proceedings of EWEA Offshore Conference, Copenhagen, 2015".

l.131: It took me a bit of time to figure where the BorWin cluster was and of which farms it consisted. Maybe you can make it clearer on the figures or the text.

l.189: Out of curiosity, shouldn't the curvature of the Earth also affect the projection of the horizontal velocity?

l.200: What is meant by the "normalization of all grid points" ? Could you precise this further?

l.210: You must have selected a reference wind speed to compute the thrust coefficient, can you mention how/where you picked this reference speed?

l.234: I'm not good with color names but I wonder if "light red" can be replaced by a different name, or the color replaced.

l.309: The streaks are quite interesting. Do you think the streaks could be related to the blockage, the wake of neighboring wind farms, or they might disappear with more samples?

l.310: The abbreviation OSS is only used here and was not introduced before.

Figure 3: I was surprised to see that the uncertainty does not increase significantly further away from the measurement. Is it because of the homogeneous assumption? Doesn't the uncertainty along the LOS increases with distance?

Figure 4: the caption mention that the conditions are "low thrust", but section 3.2 mention that the farm is not operating. This is somewhat confusing. Does the caption needs to be updated?

l.418: The upscaling assumes a profile shape. Could it be that blockage will result in

vertical profile different from the one assumed and hence affect the results?

---

## Referee Comment (RC2) · Bleeg James (Referee) · 27 Dec 2020

**Review for "Offshore wind farm global blockage measured with scanning lidar" by Jörge Schneemann et al.**

**Referee:** James Bleeg

**General comments**

The paper reports on the results of an effort to use lidar PPI scans to discern the effect of global blockage on wind speeds upstream of a large offshore wind farm. It is a welcome addition to a small club: research articles focused on field observations related to wind-farm-scale blockage. This paper, to my knowledge, is the only one so far to show in any sort of detail how the effect of global blockage varies spatially. Of additional note, the paper demonstrates that the global blockage effect is much more pronounced when the boundary layer is stable (in fact, no effect was evident when the boundary layer was unstable). The paper is well-written and includes an illuminating treatment on the uncertainties of the analysis. I consider the paper to be a significant contribution to the understanding of wind farm flows and blockage effects in particular.

Having said that, I do have a number of questions and comments about the paper. For one, the potential impact of wakes from neighboring wind farms should be discussed. These wakes could potentially influence the observed horizontal variation in wind speed and perhaps even the vertical profiles, which would have implications for the height extrapolation of the measurements. I also wonder whether more could be done to investigate blockage effects during unstable conditions and, more generally, to better target the wind speed conditions where blockage effects are expected to be most influential.

I often struggle to determine how to numerically rate a paper or to distinguish between minor revisions and major revisions. The upshot here is that this is important research that deserves publication. However, I also believe the paper would benefit from addressing the comments below.

**Specific comments**

**Potential impact of other wind farms on the results.** The BorWin wind farm cluster is 24 km to the SW of the lidar devise. The Gemini wind farm is 54 km to the SSW. These are the primary scanning directions. According to "Cluster wakes impact on a far-distant offshore wind farm's power" by Schneeman et al, these wind farms are close enough to influence the lidar measurements during stable conditions and perhaps even during weakly unstable conditions.

The objective of the measurement campaign is to isolate the impact of global blockage on the wind speeds upstream of the wind farm. If the background wind speeds (i.e. the wind speeds absent the influence of the Global Tech 1 wind farm) were horizontally homogeneous on average, then we could more confidently quantify the impact of Global Tech 1 on the horizontal variation of wind speeds as equal to the observed average horizontal variation of wind speeds.

I think the paper needs to discuss this issue and whether we should expect the background wind speeds upstream of Global Tech 1 to be horizontally homogeneous on average. I suspect that on average that the background wind speed may have a positive streamwise gradient as the wakes from neighboring wind farms recover—at least in certain meteorological conditions. The measurements in Platis et al ("First in situ evidence of wakes in the far field behind offshore wind farms" in Nature) show a good illustration of this with strong streamwise gradients 15 km to 30 km downstream of an offshore wind farm (e.g. Figure 5a). Such a situation could potentially hide/offset

blockage effects that might otherwise be observable or reduce the strength of effects that are observed.

Ideally, these questions could be answered using on-site observations; however, the lidars do not appear to reach far enough upstream or far enough back in time to assess the background flow. Nevertheless, the authors may have some data or insights that can shed light on this matter, which again, I think needs to be discussed in the paper.

**Uncertainty in profiles used to extrapolate to platform height.** The uncertainty section is very good, allowing the reader to understand exactly what is accounted for and how in the uncertainty estimates. The applicability of the log profile is listed as one of the items not accounted for—and I'm not sure how you could quantify it absent more detailed measurements at the site. But it may deserve additional comment. Might, for example, the wakes from BorWin or even Gemini cause the actual profiles to deviate from theory?

**The findings and discussion around the unstable flow results:** For unstable conditions, there was no evidence of global blockage in the presented measurements or in two additional sets of lidar measurements that the authors analyzed, but did not show in the paper. The paper concludes that the observable blockage effect is much stronger when the boundary layer is stable as compared when it is unstable. This is an important finding.

There are two related points, however, that at present I hesitate to fully embrace and perhaps more could be done to improve and/or substantiate them. The first one relates to the physical explanation for why blockage effects are so much more evident when the boundary layer is stable.

The paragraph starting on line 386 covers a lot of ground and is a good starting point for the matters I would like to discuss:

"In unstable conditions with wind speeds from 10 m s−1 to 13m s−1 and a moderate to high thrust coefficient (Scenario 1, Figure 3) we could not identify decreasing wind speeds in front of the wind farm and thus no global blockage effect. This result is plausible since wind speed fluctuations in unstable flows are much higher due to convection than the assumed magnitude of global blockage. Convection leads to more mixing in the boundary layer and thus repeals global blockage. Furthermore, 390 in unstable stratification, the boundary layer is typically higher and thus the flow can pass obstacles like hills (Stull, 1988) or in this case a wind farm more easily. Additionally to Scenario 1 we performed the analysis for unstable stratification and the wind speed ranges above rated wind speed and below cut-in wind speed respectively (c.f. Section 2.4). In both cases, we could not identify decreasing wind speeds in the inflow of the wind farm. As explained earlier we do not show these results here for brevity."

The wind speed fluctuations are indeed much higher in the unstable boundary layer, and for any given snapshot of the flow, these turbulent fluctuations will probably drown out the global blockage signal. However, on average, the adverse pressure gradient associated with Global Blockage will operate on the flow over large distances and from the ground to the top tip of the rotor and beyond, and I would expect that with enough measurements, turbulence would not, on average, completely hide the effect of this adverse pressure gradient. That said, in highly convective conditions, I could see how such a thing might occur. I'm just not sure if it occurs often. Do you have any analysis to settle this point?

The explanation regarding a higher boundary layer seems reasonable. I would add that inviscid effects related to stable stratification *within* the boundary layer also likely contributes to the pronounced wind speed decreases upstream of the wind farm during such conditions.

Although it makes sense to me that the observed global blockage effect would be more pronounced when the boundary layer is stable (it's also a finding consistent with our CFD analyses), I confess that I did not expect that blockage-related wind speed decreases would be absent from the observations during unstable conditions.

If I were to look such wind speed decreases, I might focus on a different set of conditions than those described in the above quoted paragraph. One would expect the blockage effects to be at a maximum between 4 m/s and 10 m/s, which is the only range of wind speeds not investigated when filtering for unstable conditions. Is there any particular reason for this, especially in light of the fact that the main wind speed filter for the stable cases was 7 m/s to 10 m/s? The coefficient of thrust for this turbine model drops off quite a bit between 10 m/s and 13 m/s. Moreover, the filter is based on the wind speed measured at the transition piece. The hub height wind speeds will be even higher, and thus the thrust coefficients will be even lower—and the blockage effect lower in turn.

**Filtered conditions:** More generally, have you considered widening and adjusting the filter range to better target the plateau of the Ct curve for both unstable and stable conditions? One approach might be to filter based on turbine power. More data points concentrated at the max Ct may reveal a stronger blockage-related signal and do so with less uncertainty (i.e. increase $N_r$ which seems low, and reduce SEM).

**Minor specific comments**

- Line 140: Is there any reason why you did not included data taken earlier than February 2019?
- Line 198: How do you know the mean wind direction for the scan?
- The uncertainty section is very good. I was also impressed with Figure 7 and the associated discussion as a way to address additional uncertainties.
- Line 420: "We do not know whether the global blockage effect is distributed equally over height but expect it to be most distinct in the rotor area especially at hub height." That is probably true within the induction zone of an individual turbine, but beyond a few rotor diameters of the wind farm, I do not expect this to be the case. I expect that the streamwise (adverse) pressure gradient associated with global blockage varies little between hub height and the sea surface. The wind speeds will generally be lower closer to the sea surface, and thus the percent impact of the adverse pressure gradient on wind speed should generally be larger. The difference should be small though—at least according to the one CFD simulation I just checked.
- The upstream influence of the wind farm will probably extend beyond 40 D. Whether the influence is material this far out (e.g. >0.5% in wind speed), we do not know. It could be material, and as such, the point should probably be mentioned in the paper. At the very least, the wind speed decreases for scenario 4 should be described as relative to the farthest upstream measurement—to avoid any mistaken assumptions that the decreases are relative to freestream.
- The explanation you provide for the lower wind speeds at the sides of the scan in Figure 6a is credible (see line 445-452). The wind farm is likely forcing the wind to diverge laterally around the wind farm, particularly when the boundary layer is stable. This could potentially cause the wind direction to deviate materially at the sides of the scan relative to the assumed wind direction, perhaps contributing to the trend seen in the figure. Most of us do not have a good feel for how much this could affect the results. If the wind direction is biased just 1 degree, for example, relative to expectations at the side of the scan, what

would be the impact on wind speed? It might be worth a sentence or two to quantify the effect for the reader.

**Technical corrections (that may not require corrections)**

- Line 22: The paper focused more on bias than uncertainty, though the sentence as written is not incorrect.
- Line 30: I struggled a bit with the sentence starting "The local wind speed…" The "standard onshore setup" probably refers to a power performance measurement, but you may wish to be more explicit about it. Putting the whole sentence together, I think you are saying that the wind speed measured at 2.5 D upstream is considered to be a freestream wind speed, practically beyond the influence of wind turbine blockage. You may wish to rephrase to make this sentence easier to read. Incidentally, I don't agree that 2.5 D upstream is beyond the influence of the test turbine, but I do agree that it effectively "considered" that way in the referenced IEC standard.
- Line 49: The authors write that Allaerts and Meyers, "relate the flow deceleration in front of the wind farm to pressure gradients induced by the gravity waves and not directly to global blockage." This could be a matter of taste, but I might write this differently. As Allaerts and Meyers put it in the paper, "wind-farm induced gravity waves are triggered by the upward displacement of the boundary-layer top due to flow blockage inside the farm." Inviscid effects related to stratification (i.e. gravity waves) modifies the blockage effect that would otherwise be seen in a purely neutral flow. One could therefore view these waves as an effect of blockage. Up to you on whether to make a change. I just wanted to provide another point of view.
- Line 90: Instead of "single points", you may with to consider something like "a small number of scattered points." The number of points we had at each wind farm ranged from 3 to 6. Of course, your main argument still stands.
- Line 417: Some of the measured heights in the 2018 Bleeg paper were at hub height (H), but others were at 0.7 H.

---

## Referee Comment (RC3) · Greg Poulos (Referee) · 28 Dec 2020

Schneemann et al. 2020 Off shore wind farm global blockage measured with scanning lidar (two presentation attachments also provided)

Overall, congratulations on a study that advances knowledge of wind farm-atmosphere interaction and the combined induction effect slowing the wind approaching a wind farm. Scanning Doppler lidar is an excellent choice for this application, as is dual-Doppler with carefully planned coordinated scanning strategies to pinpoint vector hub height wind speed at different distances upwind of different portions of the first row.

I think it is important to distinguish the fact that reverse pressure gradients caused by the wind farm that induce gravity waves and cause reduced wind speed upwind of the

wind farm are in fact the true atmospheric science name for "global blockage". I am not objecting to the use of the term "global blockage" and rather hoping for some recognition for confusion over physical phenomena that are already known well and the new lingo. All obstacles cause reverse pressure gradients, or block, the flow and wind farms are no exception. Wind farms are different from typical obstacles, such as mountains, buildings, trees, etc in that the wind farm actively produces thrust (reverse pressure gradient) above that of a simple obstacle (e.g. turbines that are not operating). This reverse pressure gradient upwind phenomenon, which induces gravity waves in stable atmospheric conditions, is not included in wake loss models (which are in fact being tuned to be wind farm-atmosphere interaction loss models because blockage is not wake) that have been tuned to reproduce energy losses based on the first upwind row production as a benchmark for 100% production. A benchmark expected free stream energy production value that is established suitably far upwind of the first row would alleviate the exclusion of gross energy production impacts due to the slowing of the wind and allow wind farm-atmosphere interaction effects to be included in engineering models that also include all-important wake effects.

I've attached a presentation that contains a summary of four case studies as yet unpublished, as I think you will find it interesting. Meteorological masts were in position in front of turbine rows before and after construction for long periods of time so the velocity-deficit impact could be measured relative to distant, highly-correlated unblocked, unwaked masts. We find overall total wind velocity deficits 2.5 RD upwind of the first row in the few percent range in the most affected positions; the most affected position is generally the center of the first string of turbines, upwind of more than a few downwind turbine strings.

Abstract:

Change "platform" on line 7 to "turbine hub". Change this sentence to "Our results showed a 4.5% decrease (range 2.5% to 6.5%) at XXX-m turbine hub height X.X km upwind of the wind farm with the ...". This would be more informative and concise

without the subjective use of "significant". Include a summary of the net measured global blockage among all conditions, if possible. Include lateral and string spacing in rotor diameter dimensions of the subject wind farm.

1 Introduction

Line 50: While Allaerts and Meyers do not call the gravity wave effects blockage that is likely just semantics as they are discussing the same phenomenon.

Overall, this is a good literature review and introduction.

2 Methods

2.1 Global Tech 1

Line 127: Please add the hub height of the Adwen turbines. This is important as the reader should be aware that your 24.6 m elevation measurements do not correspond to hub height, and by how much, for proper interpretation (e.g. relative to IEC standards and applicability to different circumstances. Line 131: "cluster is located 24 km southwest of GTI (see Figure 1). Figure 1: Please label the wind farms in the figure, or at least Borwin 1, Hohe See and Albatros. Figure 1b: Please indicate Hohe See and Albatros. Line 132: "were under construction within 1-5 km of GTI and our the lidar location with ..." Please discuss how hard target hits were treated in the lidar scans as regards potential effect on the outcome of your investigation of global blockage. Line 137: I recommend inserting a new Figure 2 showing a photograph of the Windcube 200S on the "transition piece" and explaining what a transition piece is – I'm not familiar with the term although I infer that you mean the junction of different tower sections. The reader would be interested in the experimental set up (height above surface, the nature obstacles on the platform, etc.). Line 139: the timeline stated here can be substituted in Line 125 in place of "first half of 2019" in Line 125 to be more specific. Line 141: I see that hub height measurements were not taken (or at least 24.6 m is indicated here). Please explain why you didn't use an elevation angle that would allow for a measurement closer to hub height (vector geometry noted as a complicating factor). Does this lower-than-hub-height result affect interpretation, or limit the study. Nygaard points out that ground-effect (surface roughness) is a consideration in the degree of blocking. Line 144: The 150 degree scan sector does not match the 210 degree or greater sector in Figure 1b. You can consider clarifying this incongruity. I believe the maximum extent of various 150 degree scans is depicted in Figure 1b but I'm not certain. Line 145: You should explain here that given the range of 0.5 km to 8 km that the experiment is able to acquire data from X RD to Y RD upwind of the wind farm which, based on XX literature, is sufficient to find unblocked free stream inflow wind speeds, without being so far away as to mistake changing atmospheric conditions for global blockage-induced wind fluctuation. NOTE: Later, in Line 193, you mention that data were only taken to 7000 m and you could clear up this inconsistency (I assume data recovery beyond 7 km range was too low and not of consequence). Lines 150-155: The discussion of extrapolation is a bit unclear. I think you are saying that you rectified all measurements to 24.6 m using a kind of extrapolation method. How was this done, by shear exponent? Lines 156-161: I believe you are saying that you calculated the Delta T portion of the RiB calculation by taking the difference between the 24.6 m temperature and reanalysis ocean surface temperature. Perhaps you could explain why you didn't monitor the SST with a sensor placed at the base of the turbine (water surface) for greater accuracy. Also, perhaps explain why you didn't place a temperature sensor further up tower (or use temperature at hub height from the turbine itself). Please describe the accuracy of the method as found in previous literature to comfort the reader regarding accuracy. Line 176: I think you should conclude this section by explaining the extrapolation method (that the so-derived (RiB etc.) logarithmic wind profile was used to correct data taken at height different than 24.6 m to 24.6 m. I'm assuming that is what was done it just isn't quite clear from the text. Line 202: Do you mean < 60% as stated on Line 192? Or is 80% a new and different threshold? Line 206: Perhaps create a table to show the matrix of possible scenarios based on stable/unstable and wind speed/thrust coefficient with the scenarios you included in your study highlighted in the table? I think you

are saying that you only looked at 4-13 m/s cases bifurcated into stable or unstable based on L. It looks like you decided to look at stable, non-operating (< 4 m/s), and stable, rated or above, cases as well. You call this a "cross check" but it isn't clear why that term applies. What do you mean by cross-check? Please include the limits of L included in stable and unstable definition in the table (or in a sentence here, if you do not include a table). Line 214: Finish the sentence, "This cross-check allowed a clearer interpretation of the results because ….(?? insert reason for the reader)". Line 216: Why not use the thrust coefficients of the Adwen turbine specifically? Lines 218 to 233: This is the crux. It is unclear why you chose different wind speed and thrust coefficient ranges for Scenarios 1 and 4 (why not use the 10-13 m/s range for both? Or 7-13 m/s?). This seems to introduce an unnecessary variable and would make comparing the stable/unstable results apples-to-oranges and hard to interpret. Why didn't you use the same wind speed and thrust coefficient ranges in Scenario 1 and Scenario 4?

Line 234: 2.5 Uncertainty estimation, thank you for including this section. I did not check every step in detail but the fact that you took the time to characterize and plot uncertainty bounds is helpful to the reader. Line 305: 3.1 Scenario 1 and Figure 3. Figure 3 shows wind speed deviation data out to ~40 rotor diameters or near 5 km (40 times 116 m RD). This is not 8km as mentioned earlier, or 7 km as mentioned later (see notes above). Can you explain why you only show to ~5 km so the reader doesn't wonder? Line 321 Scenario 2 (not operating, stable) Line 343: – this is a tricky case and the apparent mean INCREASE in wind speed closer to the first wind-facing row of the wind farm is unusual and calls into question the results. I understand that you mean to explain that due to low winds and high uncertainty that the result is meaningless – e.g. that there is no evidence of a true increase in wind approaching the wind farm. However, this is an average across 60 cases, so it is statistically robust – could it be recovery of speed from the wind farms under construction and upwind? A clear statement as to the meaning of these results in the final sentence of this section, such as, "Due to x and y and z we conclude that for low wind speed and stable conditions that there is not a material change in wind speed from -40 D to -4 D." as appropriate.

Line 344 3.4 Scenario 4 and Figure 6: It appears in Figure 6a and 6b that the wind speed is not free stream at -40 D. What does Figure 6 look like if extended to -50D or -60D? Can you find a free stream speed? Perhaps without showing the 50D or 60D plot you can simply tell the reader? What do your results suggest for the wind speed reduction at the IEC 61400-12-1 standard upwind distance of 2.5 D – larger values than 2.5% to 6.5% yes? Figure 6: Does the change in shape of the global blockage effect correspond to a particular shape, such as a parabola? The effect seems most pronounced in the direction immediately upwind perpendicular to the row in which the lidar sits? Is that correct? Would the effect be worse to the southeast toward the center of the row in which the lidar sits? Line 386: Use a label "Scenario 1" here for consistency with the body text. Line 395: "Scenario 2" Line 402: "Scenario 4" In this section it is important to describe what percent of the operating time is affected by these conditions. For example, a 5% effect for 20% of the operating time of the wind farm over it's lifetime is a much smaller impact on overall energy production (such as applied in wind energy resource assessment and with wake loss models. A 5% effect over 80% of the operational lifetime is much more significant. Thus, the fraction of time a wind farm is in stable atmospheric conditions is a key governing factor for a site-specific analysis.

It is also important to note that your findings only go to 4 D, not to the standard 2.5 D from the turbine assumed by IEC 61400-12-1 and thus the results could well be worse. Line 439 and paragraph: Indeed, onshore cases are very different meteorologically, with stable atmospheres occurring in very site specific ways, requiring site specific measurements. Another key factor is the momentum reservoir above blade tip and the stability of the air in that region, as regards wake loss recovery. Line 452: Nice discussion. The need for good onsite stability measurements is clear. Line 466: We find that blockage effects maximize at the center of a turbine string (with strings down wind) and fades to zero near the edges, based on three met towers spanning the width of a turbine string. There is acceleration immediately off the edge of the last turbine in the row. See the presentation I attached for more information.

I very much appreciate the time and effort you put into this paper and I hope the comments above are helpful.

Please also note the supplement to this comment:
https://wes.copernicus.org/preprints/wes-2020-124/wes-2020-124-RC3-supplement.pdf

**Supplement:**

[Figure]

Golden – São Paulo – Cape Town – Bangalore

[Figure]

**Paradigm Shift: Wind Farm-Atmosphere Interaction (WFAI) in the Era of Large Rotors**

**With credits to the ArcVera team**

Gregory S. Poulos, PhD, Principal Atmospheric Scientist, CEO

WINDABA 2020, Breakaway 6, Wake Effects (a misnomer)

26 October 2020 COVID-19 Virtual Cape Town, South Africa

**Key WFAI Paradigm Takeaways**

- "Wake loss" is a misnomer, WFAI is the correct term

- Large rotor turbines operate mostly outside of surface layer, in more stable air, invalidating old wake model assumptions

- Role of momentum reservoir above upper blade tip is critical

- Wind farm modification of wind flow dynamics can be a critical factor in almost any case, not in current models

- Measurements to 300-m and weather modeling required

- Understanding of WFAI energy loss is the new paradigm

[Figure]

**What can you learn from a tweet?**

[Figure]

**Tweet**

[Figure]
 **Ørsted**
@Orsted

Let's just take a moment to enjoy this amazing view of the Horns Rev 2 offshore wind farm 😍 Here, the complex flow patterns formed by the wake effects behind the wind turbines are visible due to a low-hanging fog. #Offshorewind #FridayFeeling

Photo: Henrik Krogh

2:17 AM · Jul 12, 2019 · Twitter Web Client

**Wakes are noted and clear air approaches the wind farm. Farm changes atmosphere.**

**Famous Idiom**

"Cannot see the forest for the trees"

This idiom applies to wakes (the trees) and WFAI (the forest).

[Figure]

**The big picture: WFAI, not wakes.**

It is time to recognize the full challenge and complexity of energy assessment in light of the direct impacts that operating wind farms impose on the free atmosphere into which they are inserted.

Photo: Henrik Krogh

[Figure]

**Wind Farm-Atmosphere Interaction**

**WFAI Energy Loss Definition:** That amount of turbine energy production lost from gross energy due to the insertion of a given wind farm into the free stream atmosphere.

[Figure]

**WFAI Factors**

WFAI includes wind-farm-caused wakes, flow acceleration & deceleration, and modified atmospheric dynamics.

WFAI is affected by site-specific meteorological conditions up to and above blade tip including stability/inversions, wind rose/veer, and turbulence.

WFAI is affected by layout: turbine lateral and string spacing (porosity, density, depth, footprint), terrain, ground cover, tip height, other.

*Note: Wind farms cannot be fully optimized without an accurate representation of WFAI. Knowledge of 3-d time-series of wind and stability above blade tips required to minimize risk.*

[Figure]

**WFAI includes Wakes**

**Wakes**

[Figure]

**WFAI accounts for Acceleration**

[Figure]

**Acceleration**
**Flow through and
around the wind farm**

[Figure]

**WFAI Deceleration**

[Figure]

Row 4

Row 3

Row 2

Row 1

**Combined Induction Zone Deceleration Upwind of Each Row**

**1ˢᵗ row deceleration & uplift (blockage) > 1 row width upwind**

[Figure]

**WFAI Cumulative Deceleration**

Row 1

**Deceleration from Each Row
Farm Depth Matters**

**1st row deceleration > 1
row width upwind**

[Figure]

**WFAI Flow Modification**

**Wind Farm Induced Gravity Waves, or Modification of Pre-Existing Gravity Waves**

Dependent on wind farm scale, time-varying thermodynamics, atmospheric stability, vertical structure, speed etc.

Row 4

Row 3

Row 2

Row 1

Wu shows upwind deceleration to 1/3 of wind farm depth in stable atmospheric conditions, with gravity waves.

**References:** Wu, K. L. and F. Porte-Agel, 2017: Flow adjustment inside and around large finite-size wind farms, Energies, **10**, 2164.

[Figure]

**The Validation Riddle**

- **Wake loss models and energy assessment techniques have already been tuned to get the correct P50 net energy (within uncertainty) for all phenomena, including wakes, behind the first row.**

- **Riddle: Adding more losses from gross energy calls into question prior validation, does it not? Other losses must drop if WFAI loss increases.**

- **Riddle: How could we suddenly discover that whole wind farm effects on the atmosphere are now material to wakes/WERA accuracy?**

[Figure]

**So What Changed?**

- **Most significant change** over time is in the height of wind turbines (tips +50-100 m in 10-15 years), implicit rotor diameter increase

- 50% of blade sweep above the atmospheric surface layer challenging assumptions in common wake loss models and obviating improved need to understand the momentum reservoir above blade tip

- Critical offshore due to common stable marine inversion.

- Bottom Line:  Industry should move beyond common wake loss modeling.

- The new WFAI paradigm is all encompassing with regard to the effects of a wind farm on the pre-construction free atmosphere.

[Figure]

**Large Rotor Turbines: Sweep Well Above Surface Layer**

**Atmospheric surface layer depth varies: 50-200 m days, 10-100 m nights**

*Adapted from Stull (1988)*

[Figure]

**WFAI &Wake Misnomer**

- **Tuned so-called wake loss models include much more than wake losses, including acceleration, deceleration, and other effects.**

- **If validated with 1st row as 100% production no first-row blockage and already include downwind-row blockage, gravity waves/freestream flow modification, and acceleration – more than wakes**

- **If validated with free stream meteorological tower as a proxy for 100% production, then you have a WFAI model –more than wakes**

- **Conclusion: "wake loss" is a misnomer WFAI is more accurate.**

[Figure]

**Study Method for WFAI: 4 Cases**

- Based on data from operational wind farms

- Pre-construction meteorological data from permanent met masts, and post-COD met & operational data (3)

  - Before/after wind speed ratios

- Operational data from wind farm before and after a nearby wind farm was constructed (1)

[Figure]

**Summary of Operational WFAI Studies**

- **Case 1, ~220-MW:** Brazil, four strings, 2.3-2.7 RD in-string spacing
- Unidirectional ESE wind rose
- WFAI energy loss 54% > ArcVera GTAP wakes

- **Case 2 ~150-MW:** Brazil, three strings, 1.6-2.0 RD in-string, ESE winds
- WFAI energy loss 50% > ArcVera GTAP and EV DAWM wakes

- **Case 3, ~250-MW:** Brazil, five strings with 1.5-1.7 RD in-string, ESE winds
- WFAI energy loss 56% > ArcVera GTAP and EV DAWM wakes

- **Case 4:** US wind farm with large wind farm built within 50 RD to north, 5 years later
- Wind rose: South and NW
- In south winds AEP reduced 0.4%
- **Long-distance blockage evidence**

**Cases 1-3: WFAI energy loss found to be 4-7% more than wake models. First-row blockage alone found to be 2-4% of gross energy.**

[Figure]

**Case 4: Long-Distance Farm Deceleration**

**Cartoon picture**

[Figure]

**South winds found to reduce production at Farm A**

Wakes

Upwind Farm A deceleration from Farm B 50 RD North

*Reduction in AEP chart, with axis from 0.0% to 1.2%, categories N, NE, E, SE, S, SW, W, NW; legend DAY (blue), NIGHT (orange)*

Unexpected: Farm A production impact from Farm B in south winds is slightly greater during daytime.

[Figure]

**The Future**

- WFAI modifications made to wake loss models starting in 2016 based on case studies

- Refinement over time to more accurate WFAI loss per research and publications

- Routine use of full weather physics high-resolution modeling with/without embedded turbines*

*already possible with numerical weather prediction models we now run; expensive but worth effort in high risk cases due to demonstrated energy error in current wake loss models, and consideration of long-distance wakes/blockage*

[Figure]

**Key WFAI Paradigm Takeaways**

- "Wake loss" is a misnomer, WFAI is the correct term

- Large rotor turbines operate mostly outside of surface layer, in more stable air, invalidating old wake model assumptions

- Role of momentum reservoir above upper blade tip is critical

- Wind farm modification of wind flow dynamics can be a critical factor in almost any case, not in current models

- Measurements to 300-m and weather modeling required

- Understanding of WFAI energy loss is the new paradigm

[Figure]

**A picture tells a thousand words:**
**Wind Farm Atmosphere Interaction (WFAI Losses)**

**Photograph of Horns Rev:**
*Showing that the complex interaction of a wind farm with the atmosphere is more than just wakes.*

Gravity wave induced by wind farm modifies freestream wind flow pattern

Acceleration disturbance around the side of wind farm causes mixing fog

Wake disturbance behind turbines, disguises combined induction zone blocking effect of downwind strings and causes mixing fog; note strongest front and edges of farm

String spacing   String spacing

Approximate wind direction

Acceleration disturbance around the side of wind farm causes mixing fog

Less turbulent accelerated zone between/around turbines; no fog

> 1 string spacing forward impact

Upwind of first row the combined axial induction zone (blocking) disturbance causes half-circle/parabola-shaped impact area with uplift-forced and/or mixing fog. The maximum impact is upwind more than string-to-string spacing in the center of the first row. Rapid fall off from parabola peak toward the outer edges of first wind-facing turbine string.

[Figure]

*Based on ArcVera Renewables R&D*
*Prepared by G. Poulos, May 2020*

*Photo credit: Henrik Krogh*

[Figure]

Golden – São Paulo – Cape Town – Bangalore

**Consistently providing trustworthy advantages throughout the project lifecycle**

Independent Technical Consulting
Hybrid/Storage/Wind/Solar Resource Assessment
Technology, Meteorology, and Engineering

Gregory S. Poulos, PhD, Principal Atmospheric Scientist, CEO

greg.poulos@arcvera.com, +1 303.882.2579

---

## Author Comment (AC1) · 9 Feb 2021

**Author's response to the three referee comments of the paper**

**Offshore wind farm global blockage measured with scanning lidar**

**Jörge Schneemann, Frauke Theuer, Andreas Rott, Martin Dörenkämper, and Martin Kühn**

We thank the three referees for the time and effort they put into reviewing our work and appreciate their positive and constructive feedback and comments. Please find below our answers to the comments. Referee comments are printed in *italic font* while our answers are printed in normal font.

**Referee 1: Emmanuel Branlard**

[R1C1a] *This is a thorough paper which presents measurement data of the global blockage effect using a scanning lidar. The authors have carefully accounted for uncertainties in the measurements, and found that blockage effects were mostly found in stable atmospheric configurations. I have some general and specific comments, I hope that addressing some of them can improve the paper.*

Authors response: Thank you for your positive feedback and the thorough review of our manuscript.

**General Comments:**

[R1C1] *The authors mention low and high thrust coefficients throughout the paper without defining the range that have been used.*

Authors response: We added the ranges of the thrust coefficient for the different scenarios to the manuscript.

*Some additional uncertainties may be worth mentioning:*

[R1C2] *The logarithmic law assumes an "undisturbed" atmosphere, without a wind farm. The induction may affect the profile and hence the extrapolations performed in this study. I wonder if this can have an important impact and might be a bit of a chicken and egg issue since the profile use to extrapolate might depend on the induction effect.*

Authors response: The shape of the vertical wind profile under the influence of the global blockage effect is indeed interesting. Nevertheless, it would be necessary to perform measurements of the wind profile in different distances upstream of the wind farm to analyze for changes in shape. This could be performed by using several met masts or, more practically, using dual Doppler lidar range height indicator (RHI) scans measuring virtual met masts. We did not perform such measurements and are thus not able to analyze the influence of global blockage on the wind profile. To keep the possible impact of an influenced wind profile low we chose the height of the lidar as reference. The extrapolation of wind speeds to hub height would have increased uncertainty.

[R1C3] *Uncertainty in the Ct curve (using the NREL5MW instead of manufacturer's curve).*

Authors response: We added the correct ranges of the thrust coefficient to the manuscript.

[R1C4] *The number of samples used for averaging (but it is roughly the same for all scenarios, so that shouldn't affect your conclusions).*

Authors response: We agree with your statement. The number of scans used for averaging resulted from the available data after filtering and categorizing. Since all numbers are in a similar magnitude we consider the results to be comparable. Furthermore, only the standard error of the mean (SEM) is affected by the number of scans. The propagated uncertainties are independent of the number of scans used.

[R1C5] *It was not clear to me how the thrust coefficient was computed (which wind speed is used as a reference), and I wonder if this can have an effect on the categorization of the cases. It might make sense to use the average Ct over a set of turbines close to the lidar to make sure this Ct is representative of the farm (though this introduce further issues for the determination of reference wind speed for waked turbine..).*

Authors response: We did not categorize for the thrust coefficient itself but for wind speeds at lidar height and the operational state of the wind farm. In the revised version of the manuscript we changed the categorization to the use of the nacelle wind speed instead. Please refer to the answers to question [R2C7] and [R2C8] for more detail.

[R1C6] *There is likely a relationship between the wind direction, the wind farm layout, and the blockage effect. Scenario 4 has a fairly different wind direction than the others. To be fair, given the layout, it could be expected that the blockage effect would be smaller for this wind direction. It might still be worth mentioning/investigating. Also, the wind direction fluctuations within a measurement period might affect the averaging of the flow field, and potentially reduce the observed blockage effect. Maybe the variability of the wind direction could be reported?*

Authors response: We agree that the wind direction and as a result of this the changed layout in flow direction can have an influence on the global blockage effect. Future measurement campaigns focusing on the measurement of global blockage in more detail should consider wind direction and wind direction changes in as much detail as possible. Longer measurement periods will be needed to collect enough wind data in narrow wind direction bins.

We included the wind direction variability (standard deviation of wind directions from contributing scans) in the captions of the figures. Furthermore, we found comparable wind speed deficits upstream for the same conditions as in Scenario 4 in different southwest wind direction bins. In the paper we show the most distinct case.

**Specific Comments:**

[R1C7] *l.52: I leaves this up to you, but since most of your references are fairly recent, I wonder if you'd consider replacing the one you have on that line by the older version of this paper from 2015: "E. Branlard, A. R. Meyer Forsting, Using a cylindrical vortex model to assess the induction zone in front of aligned and yawed rotors, Proceedings of EWEA Offshore Conference, Copenhagen, 2015".*

Authors response: When having the choice we prefer references from peer reviewed journals. Therefore we did not change this reference.

[R1C8] *l.131: It took me a bit of time to figure where the BorWin cluster was and of which farms it consisted. Maybe you can make it clearer on the figures or the text.*

Authors response: We added clear references on the BorWin 1 cluster in the text and in the figure's caption. We added the names of the wind farms belonging to the BorWin 1 cluster in the text.

[R1C9] *l.189: Out of curiosity, shouldn't the curvature of the Earth also affect the projection of the horizontal velocity?*

Authors response: Sure, the elevation angle changes locally over ground due to the curvature of the earth. But as for the tilt of the lidar the effect of changed measurement height is much more pronounced. Deviations in the elevation angle from the horizontal below one degree are safely neglectable since the transfer function $1/\cos()$ leads to an uncertainty factor of approx. 0.0001. Lidar tilts found here were well below 0.5 degrees and the angular difference between the laser beam and the tangential to the Earth's surface is less then 0.1 degrees for a measurement range of $10\,\mathrm{km}$.

[R1C10] *l.200: What is meant by the "normalization of all grid points" ? Could you precise this further?*

Authors response: Data measured by the lidar is transferred from the polar coordinate system of the device to a Cartesian co-ordinate system for further analysis. For each scan we determine the mean wind speed across the whole scan, i. e. considering all Cartesian grid points with valid data, and then normalise each of these grid points using that mean wind speed. So basically we just normalize each lidar scan using its mean wind speed. In our opinion, this normalisation process is described in enough detail in the manuscript.

[R1C11] *l.210: You must have selected a reference wind speed to compute the thrust coefficient, can you mention how/where you picked this reference speed?*

Authors response: We now name the correct ranges of the thrust coefficient corresponding to the different Scenarios in the manuscript.

[R1C12] *l.234: I'm not good with color names but I wonder if "light red" can be replaced by a different name, or the color replaced.*

Authors response: We replaced "light red" by "purple" and corrected the colors in the legend.

[R1C13] *l.309: The streaks are quite interesting. Do you think the streaks could be related to the blockage, the wake of neighboring wind farms, or they might disappear with more samples?*

Authors response: The streaks are indeed interesting. We do not relate them to blockage since we do not see a reason why streaks should appear in the induction zone of the wind farm. Wakes of neighbouring wind farms are not plausible as source either, since the signatures of single turbines are not expected to be visible $24\,\mathrm{km}$ (BorWin1) or more than $50\,\mathrm{km}$ (Gemini) downstream (c.f. Nygaard and Newcombe, 2018; Schneemann et al., 2020). Possible explanations could be roll convection (c.f. Etling and Brown, 1993) or wakes of ships in the construction field of the wind farm "Hohe See" visible in some of the used scans. We consider this speculation and thus don't think this should be added to the official manuscript but think its good to mention here in the discussion.

[R1C14] *l.310: The abbreviation OSS is only used here and was not introduced before.*

Authors response: We replaced "OSS" with "substation".

[R1C15] *Figure 3: I was surprised to see that the uncertainty does not increase significantly further away from the measurement. Is it because of the homogeneous assumption? Doesn't the uncertainty along the LOS increases with distance?*

Authors response: We assume that all uncertainties except the measurement height uncertainty are constant across the measurement domain. The measurement height error is directly related to the pitch and roll error and thus increases with distance to the lidar device. Also the LOS uncertainty is assumed to be constant with distance to the lidar. The uncertainty of the horizontal
120 wind speed is dependent on the difference between azimuth angle and wind direction as a consequence of the VAD algorithm (Equations (6) and (17)).

[R1C16] *Figure 4: the caption mention that the conditions are "low thrust", but section 3.2 mention that the farm is not operating. This is somewhat confusing. Does the caption needs to be updated?*
125 Authors response: We updated the caption including a statement of the not operating turbines.

[R1C17] *l.418: The upscaling assumes a profile shape. Could it be that blockage will result in vertical profile different from the one assumed and hence affect the results?*
Authors response: Yes, that is possible. However, with the information we have available we are not able to determine a more
130 accurate description of the wind speed profile. Please also refer to the answer to your second comment ([R1C2]) and the answer to the comment of referee 2 on profile uncertainty ([R2C4]).

**Referee 2: James Bleeg**
**General comments**
135 [R2C1a] *The paper reports on the results of an effort to use lidar PPI scans to discern the effect of global blockage on wind speeds upstream of a large offshore wind farm. It is a welcome addition to a small club: research articles focused onfield observations related to wind-farm-scale blockage. This paper, to my knowledge,is the only one so far to show in any sort of detail how the effect of global blockage varies spatially. Of additional note, the paper demonstrates that the global blockage effect is much more pronounced when the boundary layer is stable (in fact, no effect was evident when the boundary layer was*
140 *unstable). The paper is well-written and includes an illuminating treatment on the uncertainties of the analysis. I consider the paper to be a significant contribution to the understanding of wind farm flows and blockage effects in particular. Having said that, I do have a number of questions and comments about the paper.*
Authors response: We thank the reviewer for the positive feedback and thorough evaluation of the manuscript.

145 [R2C1] *For one, the potential impact of wakes from neighboring wind farms should be discussed. These wakes could potentially influence the observed horizontal variation in wind speed and perhaps even the vertical profiles, which would have implications for the height extrapolation of the measurements.*
*I also wonder whether more could be done to investigate blockage effects during unstable conditions and, more generally, to better target the wind speed conditions where blockage effects are expected to be most influential.*
150 Authors response: We answer these comments in detail in [R2C2], [R2C5] and [R2C7].

**Specific comments**
[R2C2] *Potential impact of other wind farms on the results. The BorWin wind farm cluster is 24 km to the SW of the lidar devise. The Gemini wind farm is 54km to the SSW. These are the primary scanning directions. According to "Cluster wakes*
155 *impact on a far-distant offshore wind farm's power" by Schneeman et al, these wind farms are close enough to influence the lidar measurements during stable conditions and perhaps even during weakly unstable conditions.*

Authors response: This is true. In Schneemann et al. (2020) we present long reaching wind farm cluster wakes influencing the inflow and the power of the Global Tech I wind farm using lidar data from the same campaign as we use in this study. Nevertheless, we do not expect cluster wakes to have negative influences on our findings of global blockage. Due to wake

160   recovery, we would expect an increase in wind speed with increasing distance from the wind farm generating the wake. Here we find the opposite with decreasing wind speeds downstream. Signatures of the single wind turbine wakes are not expected in the inflow of Global Tech I as presented in Schneemann et al. (2020). Nygaard and Newcombe (2018) showed dual Doppler radar measurements of a wind farm wake with the single wind turbine wakes smearing out less than 10 km behind the wind farm. In this manuscript we discuss the effect of cluster wakes on our global blockage analysis (starting in line 425 of the

165   original manuscript). In the revised version we added information on the disappearing signatures of single wakes after approx. 10 km and on the possible cluster wake recovery contradicting global blockage.

*[R2C3] The objective of the measurement campaign is to isolate the impact of global blockage on the wind speeds upstream of the wind farm. If the background wind speeds (i.e. the wind speeds absent the influence of the Global Tech 1 wind farm) were*

170   *horizontally homogeneous on average, then we could more confidently quantify the impact of Global Tech 1 on the horizontal variation of wind speeds as equal to the observed average horizontal variation of wind speeds.*
*I think the paper needs to discuss this issue and whether we should expect the background wind speeds upstream of Global Tech 1 to be horizontally homogeneous on average. I suspect that on average that the background wind speed may have a positive streamwise gradient as the wakes from neighboring wind farms recover—at least in certain meteorological conditions.*

175   *The measurements in Platis et al ("First in situ evidence of wakes in the far field behind offshore wind farms" in Nature) show a good illustration of this with strong streamwise gradients 15 km to 30 km downstream of an offshore wind farm (e.g. Figure 5a). Such a situation could potentially hide/offset blockage effects that might otherwise be observable or reduce the strength of effects that are observed.*
*Ideally, these questions could be answered using on-site observations; however, the lidars do not appear to reach far enough*

180   *upstream or far enough back in time to assess the background flow. Nevertheless, the authors may have some data or insights that can shed light on this matter, which again, I think needs to be discussed in the paper.*
Authors response: Thank you for this important aspect. We included statements on the contradicting effect of cluster wake recovery on global blockage measurements and spatial wind field inhomogeneity in the discussion.
"Recovery of the possible cluster wake influence with a positive wind speed gradient towards GT I could on the other hand

185   contradict the global blockage effect with a negative wind speed gradient. This would lead to a reduction of the observable global blockage effect."
"Different from wind tunnel measurements where all background conditions could be controlled, free field measurements cover numerous superimposing meteorological effects. In our case especially the background wind field without the presence of wind farms needs consideration. The small flow effects we observe here are in the order of magnitude as typical wind field

190   fluctuations locally (background turbulence) and over distances of some kilometres (spatial wind field variations). In a single lidar scan it would be not clear whether a wind speed gradient in front of a wind farm is caused by the global blockage effect or by a spatial variation in the background wind field. Our method using the average wind field of lots of lidar scans from different days allows us to average different spatial wind speed gradients of the background flow. In Scenarios 1-3 we found spatial wind speed gradients that are almost zero in average. Scenario 4 shows a negative wind speed gradient. We assume this

195    to be caused mainly by global blockage with the background turbulence and the spatial variation averaged out."

[R2C4] *Uncertainty in profiles used to extrapolate to platform height. The uncertainty section is very good, allowing the reader to understand exactly what is accounted for and how in the uncertainty estimates. The applicability of the log profile is listed as one of the items not accounted for—and I'm not sure how you could quantify it absent more detailed measurements at the site.*

200    *But it may deserve additional comment. Might, for example, the wakes from BorWin or even Gemini cause the actual profiles to deviate from theory?*

Authors response: Logarithmic wind profiles in the boundary layer are models that are mainly applicable on averaged wind fields. Deviations from this modelled behaviour must be expected. Therefore we recommend measurements of the vertical wind profile for future measurement campaigns with the intention to measure global blockage. With our data set we do not

205    have information on the wind profile and the logarithmic wind profile is the best choice we have. To keep the uncertainty in wind speed corrections low we chose lidar height as reference to keep height differences for the correction small.

The effect of far cluster wakes (and of global blockage) on the vertical wind profile is part of current research. We included a statement on this matter in the discussion.

"Furthermore, cluster wakes could have an influence on the prevailing wind profile that can not be quantified here. This is

210    subject of current research."

[R2C5] *The findings and discussion around the unstable flow results: For unstable conditions, there was no evidence of global blockage in the presented measurements or in two additional sets of lidar measurements that the authors analyzed, but did not show in the paper. The paper concludes that the observable blockage effect is much stronger when the boundary layer is stable*

215    *as compared when it is unstable. This is an important finding.*

*There are two related points, however, that at present I hesitate to fully embrace and perhaps more could be done to improve and/or substantiate them. The first one relates to the physical explanation for why blockage effects are so much more evident when the boundary layer is stable. The paragraph starting on line 386 covers a lot of ground and is a good starting point for the matters I would like to discuss:*

220    *"In unstable conditions with wind speeds from 10m s-1 to 13ms-1 and a moderate to high thrust coefficient (Scenario 1, Figure 3) we could not identify decreasing wind speeds in front of the wind farm and thus no global blockage effect. This result is plausible since wind speed fluctuations in unstable flows are much higher due to convection than the assumed magnitude of global blockage. Convection leads to more mixing in the boundary layer and thus repeals global blockage. Furthermore, in unstable stratification, the boundary layer is typically higher and thus the flow can pass obstacles like hills (Stull, 1988) or*

225    *in this case a wind farm more easily. Additionally to Scenario 1 we performed the analysis for unstable stratification and the wind speed ranges above rated wind speed and below cut-in wind speed respectively (c.f. Section 2.4). In both cases, we could not identify decreasing wind speeds in the inflow of the wind farm. As explained earlier we do not show these results here for brevity."*

*The wind speed fluctuations are indeed much higher in the unstable boundary layer, and for any given snapshot of the flow,*

230    *these turbulent fluctuations will probably drown out the global blockage signal. However, on average, the adverse pressure gradient associated with Global Blockage will operate on the flow over large distances and from the ground to the top tip of the rotor and beyond, and I would expect that with enough measurements, turbulence would not,on average, completely hide the effect of this adverse pressure gradient. That said, in highly convective conditions, I could see how such a thing might occur.*

*I'm just not sure if it occurs often. Do you have any analysis to settle this point?*

Authors response: Our physical explanation of a reduced or maybe not even present or detectable global blockage effect in unstable conditions is not related to a "noisy" measurement signal due to convection and turbulence. We assume convection to be responsible for vertical momentum transport repealing global blockage. We think it is hard to distinguish these different contributions to the inflow and suggest to study these effects in more detail using high fidelity simulations like LES. Future measurements of single situations can serve for validation. We do not have further analysis to deeper analyze these effects from our data.

We updated the text in the manuscript cited above to better describe the concept of vertical momentum transport due to convection.

[R2C6] *The explanation regarding a higher boundary layer seems reasonable. I would add that inviscid effects related to stable stratification within the boundary layer also likely contributes to the pronounced wind speed decreases upstream of the wind farm during such conditions.*

*Although it makes sense to me that the observed global blockage effect would be more pronounced when the boundary layer is stable (it's also a finding consistent with our CFD analyses), I confess that I did not expect that blockage-related wind speed decreases would be absent from the observations during unstable conditions.*

Authors response: We expect global blockage to appear in unstable stratification but we expect it to be reduced and repealed especially by vertical momentum transport due to convection. Our measurement campaign was not designed for global blockage analysis. We do not rule out the possibility that in a future refined measurement campaign global blockage (with an expected small magnitude) will be found in unstable stratification.

[R2C7] *If I were to look such wind speed decreases, I might focus on a different set of conditions than those described in the above quoted paragraph. One would expect the blockage effects to be at a maximum between 4m/s and 10 m/s, which is the only range of wind speeds not investigated when filtering for unstable conditions. Is there any particular reason for this, especially in light of the fact that the main wind speed filter for the stable cases was 7 m/s to 10 m/s? The coefficient of thrust for this turbine model drops off quite a bit between 10 m/s and 13 m/s. Moreover, the filter is based on the wind speed measured at the transition piece. The hub height wind speeds will be even higher, and thus the thrust coefficients will be even lower—and the blockage effect lower in turn.*

Authors response: You are right, this definition of scenarios was not ideal. We therefore adjusted the definition of the Scenarios with high thrust coefficients (Scenario 1 and 4). In both cases we now consider wind speeds of $8 - 11\,\mathrm{m\,s^{-1}}$ to cover the highest thrust coefficients. We further decided to use 10-minute mean wind speed measurements at the nacelle of turbine GT58 (position of the lidar) instead of lidar wind speed at TP height to select the scans for all four scenarios. In combination with the farm's power production, this should provide us with a realistic representation of the wind farms operational state.

While this adjustment of parameters slightly changes the results of all four scenarios, the conclusions we draw remain the same.

[R2C8] *Filtered conditions: More generally, have you considered widening and adjusting the filter range to better target the plateau of the Ct curve for both unstable and stable conditions? One approach might be to filter based on turbine power. More data points concentrated at the max Ct may reveal a stronger blockage-related signal and do so with less uncertainty (i.e. increase Nr which seems low, and reduce SEM).*

Authors response: Please refer to the answer of your previous question [R2C7]. We are confident that the improved wind speed intervals in combination with the wind farm and wind turbine power filter allow us to select those scans with the wind farm
275 operating at the plateau of the thrust coefficient curve.

***Minor specific comments***

[R2C9] *Line 140: Is there any reason why you did not included data taken earlier than February 2019?*
Authors response: The original aim of the measurement campaign was not to measure global blockage. Therefore, a number
280 of different scan trajectories were measured. We stared measuring the trajectories used in this work in January 2019. Further, we were restricted by the availability of SCADA data to determine the operational state of the wind farm and to estimate the tilt of the lidar device.

[R2C10] *Line 198: How do you know the mean wind direction for the scan?*
285 Authors response: We here refer to the mean wind direction $\chi$ determined using the VAD algorithm (sinusoidal fit to data points).

[R2C11] *The uncertainty section is very good. I was also impressed with Figure 7 and the associated discussion as a way to address additional uncertainties.*
290 Authors response: Thank you for the positive feedback.

[R2C12] *Line 420: "We do not know whether the global blockage effect is distributed equally over height but expect it to be most distinct in the rotor area especially at hub height." That is probably true within the induction zone of an individual turbine, but beyond a few rotor diameters of the wind farm, I do not expect this to be the case. I expect that the streamwise*
295 *(adverse) pressure gradient associated with global blockage varies little between hub height and the sea surface. The wind speeds will generally be lower closer to the sea surface, and thus the percent impact of the adverse pressure gradient on wind speed should generally be larger. The difference should be small though—at least according to the one CFD simulation I just checked.*
Authors response: You are addressing a very interesting point that needs further research in the future. We assume factors influ-
300 encing the vertical extent of global blockage to be surface roughness, the shape of the wind profile, boundary layer height, and lower blade tip height (gap between surface and lower tip height might enable higher wind speeds below the rotor area). Findings from CFD suggest a rather constant global blockage effect below hub height due to the presence of the ground (Branlard et al., 2020). Therefore we changed the statement on the expected maximum of global blockage at hub height in the manuscript to:
305 "We do not know the vertical distribution of the global blockage effect but expect it to be equally distributed between the surface and upper blade tip height. Bleeg et al. (2018) found only small variations in the analyzed blockage effect comparing mast measurements at hub height and 70 % of the hub height. CFD results indicate a rather constant global blockage effect up to at least hub height due to the presence of the ground (Branlard et al., 2020)."

310 [R2C13] *The upstream influence of the wind farm will probably extend beyond 40 D. Whether the influence is material this far out (e.g. >0.5% in wind speed), we do not know. It could be material, and as such, the point should probably be mentioned in*

*the paper. At the very least, the wind speed decreases for scenario 4 should be described as relative to the farthest upstream measurement—to avoid any mistaken assumptions that the decreases are relative to free stream.*

Authors response: We added the statement "We assume the wind speed at 40 $D$ upstream to be free stream speed but can not be sure whether small wind farm induced effects reach even further." in the discussion section.

[R2C14] *The explanation you provide for the lower wind speeds at the sides of the scan in Figure 6a is credible(see line445-452). The wind farm is likely forcing the wind to diverge laterally around the wind farm, particularly when the boundary layer is stable. This could potentially cause the wind direction to deviate materially at the sides of the scan relative to the assumed wind direction, perhaps contributing to the trend seen in the figure. Most of us do not have a good feel for how much this could affect the results. If the wind direction is biased just 1degree, for example, relative to expectations at the side of the scan, what would be the impact on wind speed? It might be worth a sentence or two to quantify the effect for the reader.*

Authors response: We added a new subsection 2.5.4 in the uncertainty estimation focusing on this issue. The newly introduced Figure 3 shows an example for the wind diverging laterally by two degrees from the mean wind direction around the wind farm. In this credible example the estimated horizontal wind speed is more than 4 % lower than the actual value at an azimuth angle difference from the wind direction of $50°$. Furthermore, we updated the Discussion section accordingly.

**Technical corrections (that may not require corrections)**

[R2C15] *Line 22: The paper focused more on bias than uncertainty, though the sentence as written is not incorrect.*

Authors response: We changed "uncertainty" to "bias".

[R2C16] *Line 30: I struggled a bit with the sentence starting "The local wind speed..." The "standard onshore setup" probably refers to a power performance measurement, but you may wish to be more explicit about it. Putting the whole sentence together, I think you are saying that the wind speed measured at 2.5 D upstream is considered to be a free stream wind speed, practically beyond the influence of wind turbine blockage. You may wish to rephrase to make this sentence easier to read. Incidentally, I don't agree that 2.5 D upstream is beyond the influence of the test turbine, but I do agree that it effectively "considered" that way in the referenced IEC standard.*

Authors response: We rephrased this sentence as follows:

"The standard for onshore power curve measurements of wind turbines recommends to measure the free wind speed, i.e. the wind speed at the turbine location in absence of the turbine, at least 2.5 rotor diameters D upstream or lateral to the turbine's location (IEC, 2017). It is assumed that the influence of a wind turbine's induction zone is very low at this distance."

[R2C17] *Line 49: The authors write that Allaerts and Meyers, "relate the flow deceleration in front of the wind farm to pressure gradients induced by the gravity waves and not directly to global blockage." This could be a matter of taste, but I might write this differently. As Allaerts and Meyers put it in the paper, "wind-farm induced gravity waves are triggered by the upward displacement of the boundary-layer top due to flow blockage inside the farm." Inviscid effects related to stratification (i.e. gravity waves) modifies the blockage effect that would otherwise be seen in a purely neutral flow. One could therefore view these waves as an effect of blockage. Up to you on whether to make a change. I just wanted to provide another point of view.*

Authors response: We agree with your formulation, which is more clear. We rephrased to: "Using LES, Allaerts and Meyers (2017) determined wind farms to excite gravity waves in stable stratification which are caused by the upward movement of

the top of the boundary-layer due to global wind farm blockage. Those gravity waves are similar to so called mountain waves induced by hills and mountains.".

[R2C18] *Line 90: Instead of "single points", you may with to consider something like "a small number of scattered points."* *The number of points we had at each wind farm ranged from 3 to 6. Of course, your main argument still stands.*
Authors response: We implemented this suggestion.

[R2C19] *Line 417: Some of the measured heights in the 2018 Bleeg paper were at hub height(H), but others were at 0.7 H.*
Authors response: Thanks for this hint. We changed the sentence to "... while Bleeg et al. (2018) used mainly measurements at hub height and some at 70 % of the hub height." and added a statement on the results of the different measurement heights in the discussion: "Bleeg et al. (2018) found only small variations in the analyzed blockage effect comparing mast measurements at hub height and 70 % of the hub height."

**Referee 3: Greg Poulos**

**General comments:**

[R3C1] *Overall, congratulations on a study that advances knowledge of wind farm-atmosphere interaction and the combined induction effect slowing the wind approaching a wind farm. Scanning Doppler lidar is an excellent choice for this application, as is dual-Doppler with carefully planned coordinated scanning strategies to pinpoint vector hub height wind speed at different distances upwind of different portions of the first row.*
*I think it is important to distinguish the fact that reverse pressure gradients caused by the wind farm that induce gravity waves and cause reduced wind speed upwind of the wind farm are in fact the true atmospheric science name for "global blockage". I am not objecting to the use of the term "global blockage" and rather hoping for some recognition for confusion over physical phenomena that are already known well and the new lingo. All obstacles cause reverse pressure gradients, or block, the flow and wind farms are no exception. Wind farms are different from typical obstacles, such as mountains, buildings, trees, etc in that the wind farm actively produces thrust (reverse pressure gradient) above that of a simple obstacle (e.g. turbines that are not operating).*
*This reverse pressure gradient upwind phenomenon, which induces gravity waves in stable atmospheric conditions, is not included in wake loss models (which are in fact being tuned to be wind farm-atmosphere interaction loss models because blockage is not wake) that have been tuned to reproduce energy losses based on the first upwind row production as a bench-mark for 100% production. A benchmark expected free stream energy production value that is established suitably far upwind of the first row would alleviate the exclusion of gross energy production impacts due to the slowing of the wind and allow wind farm-atmosphere interaction effects to be included in engineering models that also include all-important wake effects.*
Authors response: We keep the term "global blockage" throughout the paper and included these sentences in the introduction for clarification: "As for solid obstacles in the flow like mountains or buildings a wind farm represents an obstacle causing an upstream reverse pressure gradient which results in reduced wind speeds. Different from solid objects wind farms are porous and actively produce thrust. In case of a wind farm this reverse pressure gradient is referred to as global blockage."
Further we changed the definition of the term global blockage in the second last paragraph of the introduction to:
"In this paper we use the terms *blockage effect* and *wind turbine blockage effect* for decreased wind speeds in the induction

zone of single turbines while we call the accumulated blockage effect of all turbines within a wind farm or wind farm cluster, i.e. the reverse upstream pressure gradient of the wind farm, *global (wind farm) blockage* or *global (wind farm) blockage effect*."

[R3C1a] *I've attached a presentation that contains a summary of four case studies as yet unpublished, as I think you will find it interesting. Meteorological masts were in position in front of turbine rows before and after construction for long periods of time so the velocity-deficit impact could be measured relative to distant, highly-correlated unblocked, unwaked masts. We find overall total wind velocity deficits 2.5 RD upwind of the first row in the few percent range in the most affected positions; the most affected position is generally the center of the first string of turbines, upwind of more than a few downwind turbine strings.* Authors response: Thank you for sharing these interesting results. The case studies using met masts seem to be comparable to the only other study on this matter performed by Bleeg et al. (2018).

**Specific comments:**

*Abstract:*

[R3C2] *Change "platform" on line 7 to "turbine hub".*

Authors response: We did not investigate the blockage effect at turbine hub height but at the height of the turbine's transition piece. We thus changed "platform" to "transition piece".

[R3C3] *Change this sentence to "Our results showed a 4.5% decrease (range 2.5% to 6.5%) at XXX-m turbine hub height X.X km upwind of the wind farm with the . . .". This would be more informative and concise without the subjective use of "significant".*

Authors response: We implemented your suggestion on the sentence's structure.

[R3C4] *Include a summary of the net measured global blockage among all conditions, if possible.*

Authors response: We included the two missing operating conditions in the abstract:

"In contrast, at unstable stratification and similar operating conditions and for situations with low thrust coefficients we identified no wind speed deficit."

[R3C5] *Include lateral and string spacing in rotor diameter dimensions of the subject wind farm.*

Authors response: Due to the inhomogeneous layout of the wind farm Global Tech I an explanation of turbine spacings in different directions is in our opinion misleading here. The interested reader can find the layout of the wind farm in Figure 1 of the manuscript.

*1 Introduction*

[R3C6] *Line 50: While Allaerts and Meyers do not call the gravity wave effects blockage that is likely just semantics as they are discussing the same phenomenon.*

Authors response: Please see our answer to Reviewer 2 [R2C17] on this matter.

*2 Methods*
*2.1 Global Tech 1*

[R3C7] *Line 127: Please add the hub height of the Adwen turbines. This is important as the reader should be aware that your*
430 *24.6 m elevation measurements do not correspond to hub height, and by how much, for proper interpretation (e.g. relative to IEC standards and applicability to different circumstances.*
Authors response: We added the hub height in the manuscript.

[R3C8] *Line 131: "cluster is located 24 km southwest of GTI (see Figure 1). Figure 1: Please label the wind farms in the*
435 *figure, or at least Borwin 1, Hohe See and Albatros. Figure 1b: Please indicate Hohe See and Albatros.*
Authors response: We indicated the BorWin1 cluster in Figure 1 and in its caption, refer to its color and name the wind farms in this cluster in the text. We included the shapes of Hohe See and Albatros in Figure 1 b and named them in the caption.

[R3C9] *Line 132: "were under construction within 1-5 km of GTI and our the lidar location with . . ." Please discuss how hard*
440 *target hits were treated in the lidar scans as regards potential effect on the outcome of your investigation of global blockage.*
Authors response: We included the upstream distances of the Hohe See turbine locations in the manuscript and added a statement on the treatment of hard targets:
"During our measurements the wind farms «Hohe See» and «Albatros» (open blue shapes in Figure 1 (a) and turbine coordinates in Figure 1 (b)) were built in the direct south west vicinity of GTI within approx. 1 to 6 km distance upstream with
445 several transition pieces, turbines and two sub-stations installed. Measurements after the first power was fed in on 15 July 2019 were not considered (EnBW, 2019). Hard targets in the lidar data due to the construction of the two upstream wind farms «Hohe See» and «Albatros» were filtered from the measurements (c.f. Section 2.4) and can lead to a reduced data availability on the corresponding line of sight direction due to (partial) laser beam coverage."

450 [R3C10] *Line 137: I recommend inserting a new Figure 2 showing a photograph of the Windcube 200S on the "transition piece" and explaining what a transition piece is – I'm not familiar with the term although I infer that you mean the junction of different tower sections. The reader would be interested in the experimental set up (height above surface, the nature obstacles on the platform, etc.).*
Authors response: We added the term "platform to access the turbine" when introducing the transition piece (TP) and refer
455 to our previous publication (Schneemann et al., 2020) where we already showed a photograph of the lidar on the TP. Aside the wind turbine tower, some small devices like lamps on the railing, the neighbouring wind turbines and sometimes ships influence the free view of the lidar.

[R3C11] *Line 139: the timeline stated here can be substituted in Line 125 in place of "first half of 2019" in Line 125 to be*
460 *more specific.*
Authors response: We implemented this suggestion.

[R3C12] *Line 141: I see that hub height measurements were not taken (or at least 24.6 m is indicated here). Please explain why you didn't use an elevation angle that would allow for a measurement closer to hub height (vector geometry noted as a*
465 *complicating factor). Does this lower-than-hub-height result affect interpretation, or limit the study. Nygaard points out that ground-effect (surface roughness) is a consideration in the degree of blocking.*
Authors response: Global blockage analysis was not the intended goal of the measurement campaign as stated in the manuscript.

Nevertheless, during the campaign also scans with an elevation of 0.8° were performed. While these scans measure close to hub height for a certain measurement distance, the range of measurement heights will be much larger as compared to the 0° elevation scans. For the blockage analysis we were interested in measurements at a constant height throughout the measurement domain. Thus, for both elevation cases a wind speed extrapolation to a common height was required. We decided to use the horizontal 0° scans instead of the 0.8° scans in order to reduce extrapolation errors as much as possible. We decided not to mix up both elevations in order to keep scans more comparable.

We agree that the vertical extend of global blockage is very interesting and that it might be influenced by surface roughness. For a more detailed answer please refer to the question and corresponding answer of referee 2 on the vertical extend of global blockage [R2C12].

[R3C13] *Line 144: The 150 degree scan sector does not match the 210 degree or greater sector in Figure 1b. You can consider clarifying this incongruity. I believe the maximum extent of various 150 degree scans is depicted in Figure 1b but I'm not certain.*

Authors response: Yes, that's correct. Figure 1 (b) summarises four different scanning trajectories, that were chosen according to wind direction. We added a sentence to the caption of the figure for clarity.

[R3C14] *Line 145: You should explain here that given the range of 0.5 km to 8 km that the experiment is able to acquire data from X RD to Y RD upwind of the wind farm which, based on XX literature, is sufficient to find unblocked free stream inflow wind speeds, without being so far away as to mistake changing atmospheric conditions for global blockage-induced wind fluctuation. NOTE: Later, in Line 193, you mention that data were only taken to 7000 m and you could clear up this inconsistency (I assume data recovery beyond 7 km range was too low and not of consequence).*

Authors response: We added the ranges in units of rotor diameters in the Subsection "Lidar measurements". We use data up to 7000 m range to define the data availability within each scan as stated in Section 2.4 but in principle use data form the whole defined measurement range up to 7990 m. As you mention, data availability beyond 7000 m is in many cases low. This way we try to avoid neglecting scans with low data availability beyond 7000 m but otherwise satisfactory availability.

We do not want to discuss the possible spatial extent of global blockage in this place, i.e. in the Methods section, but rather in the Discussion Section. A reference clearly stating the spatial extent of global blockage is not known to us. Furthermore, we expect no clear boarder of the global blockage effect but a slow decrease until it is not detectable anymore. Please confer as well to the answer to question [R2C13].

[R3C15] *Lines 150-155: The discussion of extrapolation is a bit unclear. I think you are saying that you rectified all measurements to 24.6 m using a kind of extrapolation method. How was this done, by shear exponent?*

Authors response: We used a stability corrected logarithmic wind speed profile as stated in Equation 4 to extrapolate wind speed to a common height of 24.6 m. This is explained in Section 2.4, Line 201–202 of the revised manuscript.

[R3C16] *Lines 156-161: I believe you are saying that you calculated the Delta T portion of the RiB calculation by taking the difference between the 24.6 m temperature and reanalysis ocean surface temperature. Perhaps you could explain why you didn't monitor the SST with a sensor placed at the base of the turbine (water surface) for greater accuracy. Also, perhaps explain why you didn't place a temperature sensor further up tower (or use temperature at hub height from the turbine itself).*

*Please describe the accuracy of the method as found in previous literature to comfort the reader regarding accuracy.*

Authors response: Buoy measurements of the SST are available only for the period 9. August 2018 to 31. January 2019. The used OSTIA SST was found to be in good agreement with the buoy measurements from that period (Schneemann et al., 2020).

510 Besides that, also temperature measurements at nacelle position were available for some turbines and during some time periods. However, these sensors are not calibrated and possibly not very reliable, which is why we refrain from using them. We agree that buoy measurements or high quality temperature measurements at additional heights might be beneficial for stability estimation. In this analysis, however, we were restricted to the available data.

An additional comment on measurements of the sea surface temperature (SST) just a few centimetres below the water surface:

515 Measurements attached to the foundation of the turbine have proven not to be practical due to especially tide and waves. Buoy measurements need to be controlled in a way that the sensor does not leave the water and measures air temperature due to a tilted buoy in the tide's stream. One promising possibility is to measure SST using remote infra red sensors on the transition piece.

520 [R3C17] *Line 176: I think you should conclude this section by explaining the extrapolation method (that the so-derived (RiB etc.) logarithmic wind profile was used to correct data taken at height different than 24.6 m to 24.6 m. I'm assuming that is what was done it just isn't quite clear from the text.*

Authors response: We motivate the estimation of atmospheric stability at the beginning of Section 2.3 and further explain the extrapolation method in Section 2.4 as it is part of the Lidar data processing.

525

[R3C18] *Line 202: Do you mean < 60% as stated on Line 192? Or is 80% a new and different threshold?*

Authors response: This is a new and different threshold. We used a 60 % threshold to select scans with high data availability. That means, individual lidar scans with a data availability below 60 % were discarded (Lines 203–205 in the revised manuscript). During the further data analysis all valid lidar scans $N$ selected for a scenario were normalised and averaged.

530 Grid points, i. e. data points on the Cartesian grid on which the polar lidar data were interpolated, for which less than 80 % of the $N$ averaged scans were valid, were neglected (Line 213–215 in the revised manuscript).

[R3C19] *Line 206: Perhaps create a table to show the matrix of possible scenarios based on stable/unstable and wind speed/thrust coefficient with the scenarios you included in your study highlighted in the table? I think you are saying that*

535 *you only looked at 4-13 m/s cases bifurcated into stable or unstable based on L. It looks like you decided to look at stable, non-operating (< 4 m/s), and stable, rated or above, cases as well. You call this a "cross check" but it isn't clear why that term applies. What do you mean by cross-check? Please include the limits of L included in stable and unstable definition in the table (or in a sentence here, if you do not include a table).*

Authors response: We inserted a table following your suggestion.

540 Instead of just looking at and showing those scenarios for which we expect a global blockage effect to be visible (e. g. the wind farm operating at a high thrust coefficient and stable stratification), we decided to additionally show cases where an occurrence is less likely (e. g. the wind farm operating at a lower thrust coefficient and stable stratification) or not expected (e. g. the wind farm not operating). The fact that we do not observe a decrease of wind speed in front of the wind farm in these unlikely cases supports our hypothesis that what we observe in the "stable, high thrust coefficient"-case is in fact global blockage. If we were

545 to observe a similar effect also in those unlikely cases, it would strongly suggest that what we observe is not global blockage

but some other effect. This is what we refer to as "cross-check" in the paper. We changed the sentence in the manuscript to "This "cross-check" allowed us to better interpret the obtained results in terms of possible wind speed gradients caused by other background phenomena."

[R3C20] *Line 214: Finish the sentence, "This cross-check allowed a clearer interpretation of the results because . . ..(?? insert reason for the reader)".*

Authors response: We inserted the sentence: "This "cross-check" allowed us to better interpret the obtained results in terms of possible wind speed gradients caused by other background phenomena." Please refer to the answer to [R3C19] for further explanation of the "cross-check".

[R3C21] *Line 216: Why not use the thrust coefficients of the Adwen turbine specifically?*

Authors response: We added the ranges of the thrust coefficient corresponding to this specific turbine to the manuscript.

[R3C22] *Lines 218 to 233: This is the crux. It is unclear why you chose different wind speed and thrust coefficient ranges for Scenarios 1 and 4 (why not use the 10-13 m/s range for both? Or 7-13 m/s?). This seems to introduce an unnecessary variable and would make comparing the stable/unstable results apples-to-oranges and hard to interpret. Why didn't you use the same wind speed and thrust coefficient ranges in Scenario 1 and Scenario 4?*

Authors response: We adjusted the wind speed ranges for Scenario 1 and Scenario 4 to be better comparable. For details please refer to the question of referee 2 on the unstable scenario [R2C7] and the filter conditions [R2C8].

[R3C23] *Line 234: 2.5 Uncertainty estimation, thank you for including this section. I did not check every step in detail but the fact that you took the time to characterize and plot uncertainty bounds is helpful to the reader.*

Authors response: Thank you, we appreciate your positive feedback.

[R3C24] *Line 305: 3.1 Scenario 1 and Figure 3. Figure 3 shows wind speed deviation data out to ~40 rotor diameters or near 5 km (40 times 116 m RD). This is not 8km as mentioned earlier, or 7 km as mentioned later (see notes above). Can you explain why you only show to ~5 km so the reader doesn't wonder?*

Authors response: We chose 40 $D$ as a compromise between the maximal possible range and a range with high data availability and quality. As Figure 3 (c) indicates the number of valid scans decreases with increasing distance to the wind farm. Simultaneously the SEM increases (Figure 3 (b)). Therefore, we do not consider measurements beyond 40 D particularly valuable for our analysis. We further wanted to keep x-axis dimensions consistent across Figures. Again, 40 D poses a compromise between Figures 3 and 6, which would allow for slightly larger ranges, and Figures 4 and 5, which do not reach ranges as high as 40 D.

[R3C25] *Line 321 Scenario 2 (not operating, stable) Line 343: – this is a tricky case and the apparent mean INCREASE in wind speed closer to the first wind-facing row of the wind farm is unusual and calls into question the results. I understand that you mean to explain that due to low winds and high uncertainty that the result is meaningless – e.g. that there is no evidence of a true increase in wind approaching the wind farm. However, this is an average across 60 cases, so it is statistically robust – could it be recovery of speed from the wind farms under construction and upwind? A clear statement as to the meaning of these results in the final sentence of this section, such as, "Due to x and y and z we conclude that for low wind speed and stable*

585    *conditions that there is not a material change in wind speed from -40 D to -4 D." as appropriate.*

Authors response: We agree that this is an interesting result. However, as explained in the manuscript, even though the relative increase of wind speed of approximately $4\%$ seems quite notable, its absolute increase is very low for the mean wind speed of $3.19\,\mathrm{m\,s^{-1}}$. Further, all of the three estimated uncertainty ranges, including the SEM, which yields information regarding the statistical significance of the results, could account for the observed increase in wind speed.

590    Please note, that with changing the filtering conditions (c. f. answer to [R2C7]) the results change slightly. Instead of the increase of wind speed previously observed for Scenario 2, we now observe a slight increase of approximately $3\%$ up to $-30\,D$ followed by a decrease of wind speed. Again, the variations of wind speed are low in absolute terms and could be explained by measurement uncertainties as shown by the three uncertainty ranges. In this figure two distinct wakes, likely caused by ships or jack-up barges used for the construction of the neighbouring wind farms are visible on both sides of the

595    wind field-cut.

Following your suggestion we added the following sentence to the end of Section 3.2: "All shown uncertainty ranges are able to account for the observed variations in wind speed. Taking into account also its aforementioned low absolute values, we consider these variations to be insignificant."

600    [R3C26] *Line 344 3.4 Scenario 4 and Figure 6: It appears in Figure 6a and 6b that the wind speed is not free stream at -40 D. What does Figure 6 look like if extended to -50D or -60D? Can you find a free stream speed? Perhaps without showing the 50D or 60D plot you can simply tell the reader? What do your results suggest for the wind speed reduction at the IEC 61400-12-1 standard upwind distance of 2.5 D – larger values than 2.5% to 6.5% yes?*

Authors response: For Scenario 4 we are restricted to a maximum range of $-44D$. From $-40D$ to $-44D$ the normalized

605    wind speed stays approximately constant at a value of 1.03. We added this information to the description of Scenario 4 in the manuscript.

The shape of the plot suggests that wind speed will further decrease when approaching the wind farm. However, we do not have measurements to confirm this assumption and can thus not make a definite statement regarding the magnitude of the decrease. Please also confer to the answer to your comment [R3C29].

610

[R3C27] *Figure 6: Does the change in shape of the global blockage effect correspond to a particular shape, such as a parabola? The effect seems most pronounced in the direction immediately upwind perpendicular to the row in which the lidar sits? Is that correct? Would the effect be worse to the southeast toward the center of the row in which the lidar sits?*

Authors response: This is a very interesting question and one that requires further investigation. In RANS simulations per-

615    formed by Bleeg et al. (2018) the shape of the blockage effect follows that of the first row of turbines (see also our discussion in Section 4.1, Line 505 ff. in the revised manuscript). Our results suggest a similar shape in the center of the lidar scan an deviate from that at the edges of the scan as you mentioned correctly. Please confer also to the answer to comment [R2C14].

[R3C28] *Line 386: Use a label "Scenario 1" here for consistency with the body text.*

620    Authors response: The label "Scenario 1" is already stated at the end of line 386.

[R3C29] *Line 395: "Scenario 2" Line 402: "Scenario 4" In this section it is important to describe what percent of the oper- ating time is affected by these conditions. For example, a 5% effect for 20% of the operating time of the wind farm over it's*

*lifetime is a much smaller impact on overall energy production (such as applied in wind energy resource assessment and with*
*wake loss models. A 5% effect over 80% of the operational lifetime is much more significant. Thus, the fraction of time a wind*
*farm is in stable atmospheric conditions is a key governing factor for a site-specific analysis. It is also important to note that*
*your findings only go to 4 D, not to the standard 2.5 D from the turbine assumed by IEC 61400-12-1 and thus the results could*
*well be worse.*

Authors response: We agree that the impact of global blockage on the annual energy production will be strongly related to the
atmospheric stability on site. We mentioned this in our discussion: "To assess the impact of the global blockage effect on a
wind farms annual energy production (AEP) more research and development on the implementation and validation of the effect
in wind farm planning tools is needed. A detailed AEP assessment then needs to consider particularly the local undisturbed
wind speed and stability wind roses." However, based on the available data we are not able to draw more detailed conclusions
on the impact of global blockage on the AEP of the wind farm Global Tech I. A stability wind rose for the Fino 1 site roughly
40 km away is given e.g. by Platis et al. (2018).

We added the following text to the discussion to account for the possible stronger global blockage effect when looking especially closer to the wind farm:

"When discussing the strength of global blockage in our data, we need to consider the measurement distances. We analyzed
a wind speed difference between 40 D and 4 D upstream. On the far distance the effect seems to have almost vanished with
a constant wind speed. Nevertheless, a further slight increase in wind speed for larger distances is possible. Moreover, the
strong wind speed gradients at the lower distance of 4 D suggest an even further wind speed decrease towards the wind farm.
Therefore, we assume the global blockage effect to be even stronger than quantified here."

[R3C30] *Line 439 and paragraph: Indeed, onshore cases are very different meteorologically, with stable atmospheres occur-*
*ring in very site specific ways, requiring site specific measurements. Another key factor is the momentum reservoir above blade*
*tip and the stability of the air in that region, as regards wake loss recovery.*
Authors response: Thank you for this interesting remark.

[R3C31] *Line 452: Nice discussion. The need for good onsite stability measurements is clear.*
Authors response: Thank you for the positive feedback.

[R3C32] *Line 466: We find that blockage effects maximize at the center of a turbine string (with strings down wind) and fades*
*to zero near the edges, based on three met towers spanning the width of a turbine string. There is acceleration immediately off*
*the edge of the last turbine in the row. See the presentation I attached for more information.*
Authors response: Thank you for providing this interesting presentation.

**References**

Allaerts, D. and Meyers, J.: Gravity Waves and Wind-Farm Efficiency in Neutral and Stable Conditions, Boundary-Layer Meteorology, 166, 269–299, https://doi.org/10.1007/s10546-017-0307-5, 2017.

Bleeg, J., Purcell, M., Ruisi, R., and Traiger, E.: Wind Farm Blockage and the Consequences of Neglecting Its Impact on Energy Production, Energies, 11, 1609, https://doi.org/10.3390/en11061609, 2018.

Branlard, E., Quon, E., Meyer Forsting, A. R., King, J., and Moriarty, P.: Wind farm blockage effects: comparison of different engineering models, Journal of Physics: Conference Series, 1618, 062 036, https://doi.org/10.1088/1742-6596/1618/6/062036, 2020.

EnBW: EnBW Hohe See and Albatros wind farms, The construction diary for Hohe See, online, https://www.enbw.com/renewable-energy/wind-energy/our-offshore-wind-farms/hohe-see/construction-diary.html, last access 27 October 2020, 2019.

Etling, D. and Brown, R. A.: Roll vortices in the planetary boundary layer: A review, Boundary-Layer Meteorology, 65, 215–248, https://doi.org/10.1007/bf00705527, 1993.

IEC: IEC 61400-12 Wind turbine generator systems – Part 12: Wind turbine power performance testing, 2017.

Nygaard, N. G. and Newcombe, A. C.: Wake behind an offshore wind farm observed with dual-Doppler radars, Journal of Physics: Conference Series, 1037, 072 008, https://doi.org/10.1088/1742-6596/1037/7/072008, 2018.

Platis, A., Siedersleben, S. K., Bange, J., Lampert, A., Bärfuss, K., Hankers, R., Cañadillas, B., Foreman, R., Schulz-Stellenfleth, J., Djath, B., Neumann, T., and Emeis, S.: First in situ evidence of wakes in the far field behind offshore wind farms, Scientific Reports, 8, https://doi.org/10.1038/s41598-018-20389-y, 2018.

Schneemann, J., Rott, A., Dörenkämper, M., Steinfeld, G., and Kühn, M.: Cluster wakes impact on a far-distant offshore wind farm's power, Wind Energy Science, 5, 29–49, https://doi.org/10.5194/wes-5-29-2020, 2020.